

**Diel variation of mercury stable isotope ratios record photoreduction of PM2.5-bound mercury**

Qiang Huang[1,2], Jiubin Chen[1,3,*], Weilin Huang[4], John R. Reinfelder[4], Pingqing Fu[5], Shengliu

5    Yuan[1], Zhongwei Wang[1], Wei Yuan[1], Hongming Cai[1], Hong Ren[5], Yele Sun[5] and Li He[6]

[1] SKLEG, Institute of Geochemistry, CAS, Guiyang 550081, China

[2] SKLOG, Guangzhou Institute of Geochemistry, CAS, Guangzhou 510640, China

[3] Institute of Surface-Earth System Science, Tianjin University, 300072, China

[4] Department of Environmental Sciences, Rutgers, The State University of New Jersey, New

10   Brunswick, NJ 08901, USA

[5] LAPC, Institute of Atmospheric Physics, CAS, Beijing 100029, China

[6] Laboratory of Hebei Institute of Regional Geology and Mineral Resources Survey,

Shijiazhuang 065000, China

*Corresponding author.

15   E-mail: jbchen@tju.edu.cn



**Abstract**. Mercury (Hg) bound to fine aerosols (PM$_{2.5}$-Hg) may undergo photochemical reaction that causes isotopic fractionation and obscures the initial isotopic signatures. In this study, we quantified Hg isotopic compositions for 56 PM$_{2.5}$ samples collected between Sept. 15th and Oct. 16th, 2015 from Beijing, China, among which 26 were collected during the daytime (between 8:00 a.m. and 6:30 p.m.) and 30 during night (between 7:00 p.m. and 7:30 a.m.). The results show that diel variation was statistically significant ($p < 0.05$) for Hg content, $\Delta^{199}$Hg and $\Delta^{200}$Hg, with Hg content during the daytime ($0.32 \pm 0.14$ µg g$^{-1}$) lower than at night ($0.48 \pm 0.24$ µg g$^{-1}$) and $\Delta^{199}$Hg and $\Delta^{200}$Hg values during the daytime (mean of 0.26‰ ± 0.40‰ and 0.09‰ ± 0.06‰, respectively) higher than during the nighttime (0.04‰ ± 0.22‰ and 0.06‰ ± 0.05‰, respectively), whereas PM$_{2.5}$ concentrations and $\delta^{202}$Hg values showed insignificant ($p > 0.05$) diel variation. Geochemical characteristics of the samples and the air mass backward trajectories (PM$_{2.5}$ source related) suggest that diel variation in $\Delta^{199}$Hg values resulted primarily from the photochemical reduction of divalent PM$_{2.5}$-Hg, rather than variations in emission sources. The importance of photoreduction is supported by the strong correlations between $\Delta^{199}$Hg and: (i) $\Delta^{201}$Hg (positive, slope = 1.1), (ii) $\delta^{202}$Hg (positive, slope = 1.15), (iii) content of Hg in PM$_{2.5}$ (negative), (iv) sunshine durations (positive), and (v) ozone concentration (positive) observed for consecutive day-night paired samples. Our results provide isotopic evidence that local, daily photochemical reduction of divalent Hg is of critical importance to the fate of PM$_{2.5}$-Hg in urban atmospheres and that, in addition to variation in sources, photochemical reduction appears to be an important process that affects both the particle mass-specific abundance and isotopic composition of PM$_{2.5}$-Hg.

*Keywords*: Mass-independent fractionation; Atmospheric particulate mercury; Transport and transformation; Isotopic evidence of photoreduction



## 1   Introduction

Atmospheric mercury (Hg) consists of three operationally-defined forms including particle-bound Hg (PBM), gaseous oxidized Hg (GOM), and gaseous elemental Hg (GEM) (Selin, 2009). GEM is the most abundant (about 90%) and chemically stable form (Selin, 2009), and is transported at regional and global scales. GOM has short residence times as it can readily be dissolved in rain droplets, adsorbed on particulate matter (PM), and it reacts rapidly within both gaseous and aqueous phases with or without PM. PBM contains mainly reactive Hg species such as $Hg^{2+}$ and perhaps trace quantities of $Hg^0$, and is transported at regional or local scales thereby reflecting Hg pollution and cycling within short distances from emission source (Selin, 2009; Subir et al., 2012). PBM has multiple sources and undergoes complex transport and transformation processes in the atmosphere (Subir et al., 2012).

This study aimed at characterizing the isotope compositions of $PM_{2.5}$-Hg (particulate matter with aerodynamic diameter less than 2.5 micrometers) to better understand the complex transformation processes of PBM. Hg has seven stable isotopes and is known to exhibit both mass-dependent (MDF, represented by $\delta^{202}Hg$) and mass-independent (MIF, including odd-mass-number isotopic MIF (odd-MIF) and even-mass-number isotopic MIF (even-MIF), represented by $\Delta^{199}Hg$, $\Delta^{201}Hg$, $\Delta^{200}Hg$ and $\Delta^{204}Hg$) fractionation during Hg transformations under various environmental conditions (Hintelmann and Lu, 2003; Jackson et al., 2004; Bergquist and Blum, 2007; Jackson et al., 2008; Gratz et al., 2010; Chen et al., 2012; Sherman et al., 2012). Prior studies have shown that MDF can be induced by several Hg transformation and transport processes (Bergquist and Blum, 2007; Kritee et al., 2007; Estrade et al., 2009; Yang and Sturgeon, 2009; Sherman et al., 2010; Wiederhold et al., 2010; Ghosh et al., 2013; Smith et al., 2015; Janssen et al., 2016), but large extents of odd-MIF mainly occurred during photochemical reactions including photoreduction (Bergquist and Blum, 2007; Zheng and Hintelmann, 2009; Sherman et al., 2010; Zheng and Hintelmann, 2010b; Rose et al., 2015),





photodemethylation (Bergquist and Blum, 2007; Rose et al., 2015) and photooxidation (Sun et

al., 2016). Smaller but measureable degrees of odd-MIF were also reported for

nonphotochemical abiotic reduction (Zheng and Hintelmann, 2010a) and evaporation of $Hg^0$

(Estrade et al., 2009; Ghosh et al., 2013). Interestingly, the results of laboratory and field

investigations suggest that specific $\Delta^{199}Hg/\Delta^{201}Hg$ ratios are associated with such

transformation processes, with a ratio of about 1.0 for photoreduction of inorganic $Hg^{2+}$

(Bergquist and Blum, 2007; Zheng and Hintelmann, 2009; Sherman et al., 2010; Zheng and

Hintelmann, 2010b; Rose et al., 2015), about 1.3 for photodemethylation and 1.6 for $Hg^0$

evaporation and photooxidation (Bergquist and Blum, 2007; Estrade et al., 2009; Zheng and

Hintelmann, 2010a; Ghosh et al., 2013; Rose et al., 2015; Sun et al., 2016). Even-MIF of Hg

isotope signatures were observed mostly in wet deposition, but the mechanism producing such

fractionation remains unknown (Chen et al., 2012; Cai and Chen, 2016).

Prior studies have shown relatively constant Hg isotope compositions for GEM and very

large variations of Hg isotope ratios for dissolved $Hg^{2+}$ in wet precipitation (Gratz et al., 2010;

Chen et al., 2012; Rolison et al., 2013; Wang et al., 2015; Yuan et al., 2015). A few studies

reported that the Hg isotope compositions of PBM also show large variations (Rolison et al.,

2013; Das et al., 2016; Huang et al., 2016; Yu et al., 2016; Xu et al., 2017). Among these limited

studies, Rolison et al. (2013) reported $\delta^{202}Hg$ (−1.61‰ to −0.12‰) and $\Delta^{199}Hg$ values (0.36‰

to 1.36‰) for Hg bound on total suspended particulates (TSP), with $\Delta^{199}Hg/\Delta^{201}Hg$ ratios of

approximately unity, a value typical of photoreduction of inorganic $Hg^{2+}$. Das et al. (2016)

found values of $\Delta^{199}Hg$ varied between −0.31‰ and 0.33‰ for $PM_{10}$ from Kolkata, eastern

India. It was suggested that PBM with longer residence times may have undergone greater

photoreduction and hence exhibited more positive MIF. Huang et al. (2016) investigated Hg

isotope compositions for $PM_{2.5}$ samples taken from Beijing, China, and attributed their

observed seasonal variations in both MDF ($\delta^{202}Hg$ from −2.18‰ to 0.51‰) and MIF ($\Delta^{199}Hg$



from −0.53‰ to 0.57‰) to varied contributions from multiple sources of PM$_{2.5}$-Hg, while the
more positive Δ$^{199}$Hg values were likely produced by extensive photochemical reduction during
long-range-transported. These prior results show that the Hg isotope approach can be employed
for tracking sources and identifying possible transformation processes for airborne PM-Hg, and
that PBM may undergo photochemical reactions that obscure its initial isotopic signature.

The goal of this study was to quantify short-term (diel) variations in the isotope composition
of PM$_{2.5}$-Hg in an effort to elucidate if photochemical processes could impact overall contents
and isotope compositions of PM-bound Hg in an urban environment. Unlike prior studies in
which PM samples were collected continuously over 24 hrs or longer, we collected two PM$_{2.5}$
samples per 24 hrs with a daytime (D) sample between 8:00 a.m. and 6:30 p.m. and a nighttime
(N) sample between 7:00 p.m. and 7:30 a.m.. It is intuitive that, while both D and N PM$_{2.5}$
samples may have similar local or regional sources if the wind trajectory remains unchanged,
D samples could have been exposed to more solar radiation than N samples, likely resulting in
diel variations in the Hg isotope compositions that are indicative of differences in
photochemical transformation of PM$_{2.5}$-Hg. The specific objectives of this study were to verify
and quantify whether Hg isotope compositions of PM$_{2.5}$ exhibit diel variations, and to elucidate
whether photochemical transformation is the dominant process for such diel variations.

## 2   Experimental section

### 2.1   Field site, sampling method, and preconcentration of PM$_{2.5}$-Hg

Beijing was selected as the area of study because of its well-known air pollution issue (Zhang
et al., 2007). Detailed information on the study site and PM$_{2.5}$ sampling procedures were given
elsewhere (Huang et al., 2016). During the sampling period between Sept. 15$^{th}$ and Oct. 16$^{th}$,
2015, the average outdoor temperatures were 22.1 ± 3.0°C and 18.5 ± 2.7°C, and the average
relative humidity were 45 ± 20% and 59 ± 19%, for D and N, respectively. The PM$_{2.5}$ samples



were collected using a Tisch Environmental PM$_{2.5}$ high volume air sampler, which collects

particles at a flow rate of 1.0 m$^3$ min$^{-1}$ through a PM$_{2.5}$ size selective inlet on a pre-combusted

(450°C for 6 hrs) quartz fiber filter (Pallflex 2500 QAT-UP, 20 cm × 25 cm, Pallflex Product

Co., USA). Quartz fiber filters were wildly used to collect operationally defined PBM

(Schleicher et al., 2015; Zhang et al., 2015; Xu et al., 2017). A total of 61 samples including 30

D samples and 31 N samples were collected between 8:00 a.m. and 6:30 p.m. and 7:00 p.m. to

7:30 a.m., respectively, along with 2 field blanks. They were wrapped with aluminum film,

packed in plastic bags, and stored at −20°C in the lab prior to analysis. Meteorological data,

including temperature (T), relative humidity (RH), sunshine duration, daily average wind speed

(WS), were acquired from China Meteorological Administration (http://data.cma.cn), and the

atmospheric ozone content ($P_{O3}$) was measured concurrently. These data are summarized in

Table S1.

### 2.2 Hg content and stable isotope measurements

The mass of each PM$_{2.5}$ sample was gravimetrically quantified. Hg bound on each PM$_{2.5}$ sample

was extracted and concentrated for analysis of Hg content and stable Hg isotopes using the

method reported previously (Huang et al., 2015). The details of the procedures are also given

in supplementary material (SI).

Among the 61 PM$_{2.5}$ samples, 56 (including 26 D- and 30 N-samples) had sufficient Hg mass

(> 10 ng) and were further analyzed for Hg isotope compositions using a multicollector

inductively coupled plasma mass spectrometer (MC-ICP-MS, Nu Instruments Ltd., UK)

equipped with a continuous flow cold vapor generation system. Detailed protocols for the Hg

isotope analysis can be found in Huang et al. (2015) and also in SI. $^{196}$Hg and $^{204}$Hg were not

measured due to their very low abundance. Instrumental mass bias was corrected using an

internal standard (NIST SRM 997 Tl) and strict sample-standard bracketing with NIST SRM





3133 Hg standard. Delta ($\delta$) notation is used to represent MDF in units of per mil (‰) as defined

by the following equation (Blum and Bergquist, 2007):

140 $\qquad \delta^x Hg\ (‰) = [(^xHg/^{198}Hg)_{sample}/(^xHg/^{198}Hg)_{NIST3133} - 1] \times 1000$ (1)

where x = 199, 200, 201, and 202. MIF is reported as the deviation of a measured delta value

from the theoretically predicted MDF value according to the equation:

$\qquad \Delta^x Hg\ (‰) = \delta^x Hg - \beta \times \delta^{202}Hg$ (2)

where the mass-dependent scaling factor $\beta$ is 0.252, 0.5024, and 0.752 for $^{199}$Hg, $^{200}$Hg and

145 $^{201}$Hg, respectively (Blum and Bergquist, 2007).

For quality assurance and control, we used NIST SRM 3177 Hg as a secondary standard and

analyzed repeatedly during sample analysis session. The collective measurements of the NIST

3177 standard yielded average $\delta^{202}$Hg, $\Delta^{199}$Hg, $\Delta^{200}$Hg and $\Delta^{201}$Hg values of −0.53‰ ± 0.09‰,

−0.01‰ ± 0.04‰, 0.00‰ ± 0.04‰ and −0.01‰ ± 0.07‰ (2SD, $n$ = 17). We also analyzed

regularly a well-known reference material UM-Almaden and a certified reference material

(CRM) GBW07405, and the results showed average $\delta^{202}$Hg, $\Delta^{199}$Hg, $\Delta^{200}$Hg and $\Delta^{201}$Hg values

of −0.60‰ ± 0.09‰, −0.01‰ ± 0.04‰, 0.01‰ ± 0.04‰ and −0.03‰ ± 0.07‰ (2SD, $n$ = 17)

and of −1.77‰ ± 0.14‰, −0.29‰ ± 0.06‰, 0.00‰ ± 0.04‰ and −0.32‰ ± 0.07‰ (2SD, $n$ =

6), respectively. These values were consistent with previous results (Blum and Bergquist, 2007;

Chen et al., 2010; Huang et al., 2015; Huang et al., 2016). The uncertainties of PM$_{2.5}$-Hg isotope

ratios listed in Table S2 were calculated based on repetitive measurements. However, if

uncertainty of the isotopic compositions for a given sample was smaller than the uncertainty of

CRM GBW07405, the uncertainty associated with that sample was assigned 2SD uncertainties

(0.14‰, 0.06‰, 0.04‰ and 0.07‰ for $\delta^{202}$Hg, $\Delta^{199}$Hg, $\Delta^{200}$Hg and $\Delta^{201}$Hg) obtained for long-

term measurement of the CRM GBW07405.

### 2.3 Air mass backward trajectories


To identify possible pathways of PM$_{2.5}$-Hg transport, backward HYSPLIT trajectories of air masses at a height of 500 m above ground level and arriving at the sampling site were simulated. Backward trajectories for each D or N sample were calculated every 1 hrs using the Internet-

Based HYSPLIT Trajectory Model and gridded meteorological data (Global Data Assimilation System, GDAS1) from the U.S. National Oceanic and Atmospheric Administration (NOAA) (Fig. S1). The obtained average directions of arriving air masses for each sample were summarized in Table S1. The frequencies of backward trajectories were calculated for all the samples taken during Sept. 15$^{th}$ to Oct. 16$^{th}$ 2015 using the Internet-Based HYSPLIT Trajectory

Model and the archived GDAS0p5, with an interval of 3 hrs. Each trajectory had a total run time of 24 hrs and a grid resolution of 0.5 × 0.5 degree trajectory frequency. The simulation results showed the dominant air mass was arriving from southwest of the sampling site during the sampling period (see Fig. 1).

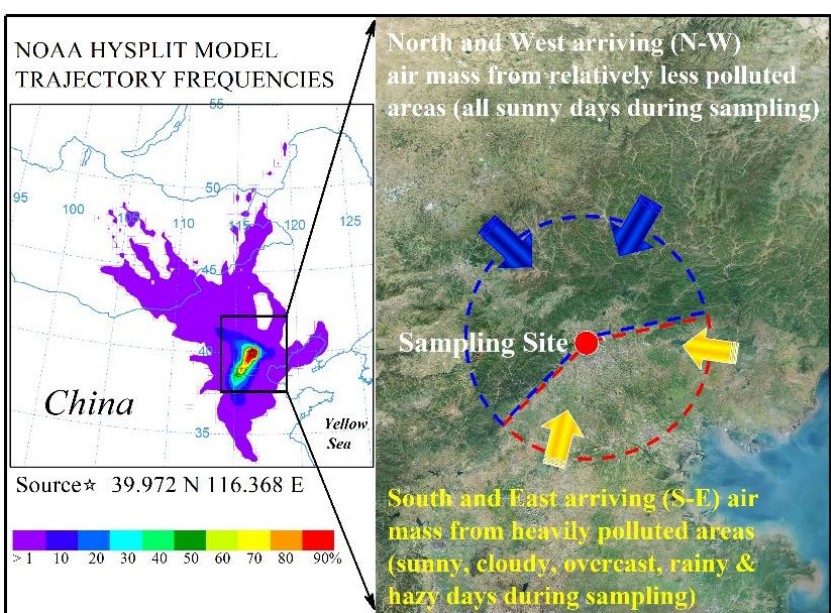

**Figure 1.** Geographic location of the PM$_{2.5}$ collection site in Beijing, China (Baidu Map image) and average air mass back trajectories during sampling from Sept. 15$^{th}$ to Oct. 16$^{th}$, 2015 (left), and characteristics of North-West vs. South-East arriving air masses.



### 2.4 Statistical analysis

T-test was performed for uncertainty analysis using IBM SPSS Statistics Version 22. Both
Paired Samples T-test and Independent Samples T-test were performed for diel variations of
Hg content, $\delta^{202}$Hg, $\Delta^{199}$Hg and $\Delta^{200}$Hg, and their results were summarized in Table S3.

### 3 Results and discussion

### 3.1 Diel variation of PM$_{2.5}$-Hg

The chronological sequence of Hg stable isotope ratios, along with PM$_{2.5}$ sample properties and
weather conditions for the 56 PM$_{2.5}$ samples are presented in Figure 2 (see also Tables S1 and
S2 for quantitative atmospheric data and $\Delta^{201}$Hg values). The major features of this dataset
include: i) large variations in both MDF and odd-MIF of Hg isotopes, ii) significant diel
differences in Hg isotope ratios, iii) correlations of weather conditions and air mass backward
trajectories with Hg isotope signals, and iv) detectable even-MIF.



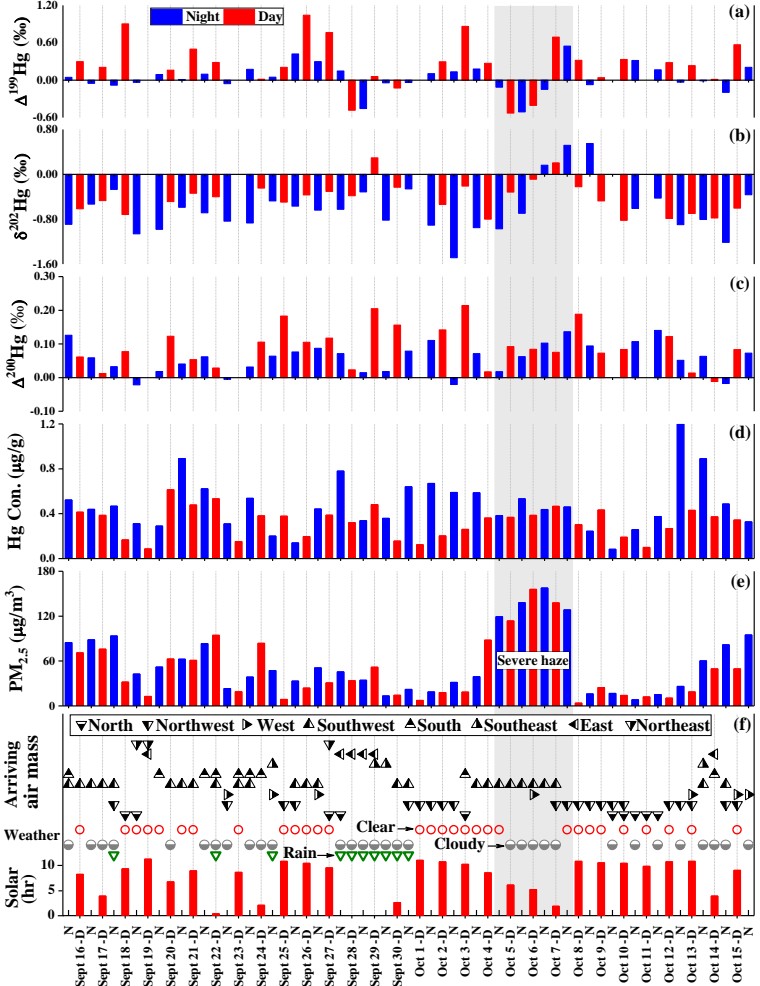

**Figure 2.** Chronological sequence of MIF ($\Delta^{199}$Hg and $\Delta^{200}$Hg), MDF ($\delta^{202}$Hg), PM$_{2.5}$-Hg contents ($C_{Hg}$), and PM$_{2.5}$ concentrations of the 56 samples collected during the daytime (D, red) and nighttime (N, blue), along with selected weather data including cumulative hours of sunshine (Solar) and air mass backward-trajectory directions.

The volumetric PM$_{2.5}$ concentrations ranged from 4 to 158 μg m$^{-3}$ with an average value of 52 ± 40 μg m$^{-3}$ (1SD, $n = 61$) and the highest values (> 100 μg m$^{-3}$) detected during a severe haze event of Oct. 4$^{th}$-7$^{th}$. The mass-based Hg contents ranged from 0.08 to 1.22 μg g$^{-1}$ with a mean value of 0.40 ± 0.21 μg g$^{-1}$ (1SD, $n = 61$).



Hg isotope analysis showed that $\delta^{202}$Hg values varied from −1.49‰ to 0.55‰ (mean = −0.53‰ ± 0.40‰, 1SD, $n = 56$), with the lowest value found in sample Oct-2-N and the highest in Oct-8-N. Significant odd-MIF of Hg isotopes was found and the $\Delta^{199}$Hg values ranged from −0.53‰ to 1.04‰ (mean = 0.14‰ ± 0.33‰) with the lowest (−0.53‰) $\Delta^{199}$Hg value on Oct-5-D during

the severe haze event and the highest (1.04‰) $\Delta^{199}$Hg value on Sept-26-D during a sunny day (without cloud) (Fig. 2). All samples also displayed slight even-MIF, with $\Delta^{200}$Hg values ranging from −0.02‰ to 0.21‰ (average 0.07‰ ± 0.06‰, 1SD, $n = 56$), which were significant compared to the detection precision of ± 0.04‰. The overall variations of Hg isotope ratios for these 12-hr D/N PM$_{2.5}$ samples are generally consistent with several prior reports for the 24-hr

PBM samples (Rolison et al., 2013; Das et al., 2016; Huang et al., 2016).

T-test results (Table S3) showed that diel variation was statistically significant ($p < 0.05$) for Hg contents, $\Delta^{199}$Hg, and $\Delta^{200}$Hg values, as Hg contents for D-samples (0.32 ± 0.14 µg g$^{-1}$) were lower than N-samples (0.48 ± 0.24 µg g$^{-1}$), and $\Delta^{199}$Hg and $\Delta^{200}$Hg values for D-samples (mean of 0.26‰ ±0.40‰ and 0.09‰ ± 0.06‰, respectively) were higher than N-samples

(−0.04‰ ± 0.22‰ and 0.06‰ ± 0.05‰, respectively). However, PM$_{2.5}$ concentrations and $\delta^{202}$Hg had statistically insignificant ($p > 0.05$) diel variation.

### 3.2    Diel variation in odd-MIF of PM$_{2.5}$-Hg independent of air mass source

Many consecutive D-N sampling intervals had similar air mass back trajectories (Table S1 and

Fig. S1), suggesting that the dominant sources of PM$_{2.5}$-Hg did not vary over each such 24 hr sampling period. For example, pairs Sept-16-D and Sept-16-N, Sept-17-D and Sept-17-N, Sept-20-D and Sept-20-N, Sept-21-D and Sept-21-N, Oct-1-D and Oct-1-N, Oct-2-D and Oct-2-N, Oct-4-D and Oct-4-N have similar air mass trajectories from the southwest, and pairs Oct-8-D and Oct-8-N, Oct-9-D and Oct-9-N, Oct-10-D and Oct-10-N, Oct-11-D and Oct-11-N, Oct-12-

D and Oct-12-N have similar air mass trajectories from the northwest and north (Fig. S1). It is





reasonable to assume, therefore, that each of these D-N PM$_{2.5}$ sample pairs had identical dominant sources of PM$_{2.5}$-Hg and to expect that they would have very similar Hg isotope compositions. Instead, however, the data presented in Table S2 and Figure 2 revealed a unique and consistent pattern of diel variation in Hg isotope ratios; specifically, each PM$_{2.5}$ D-sample

had a statistically significantly higher positive $\Delta^{199}$Hg value (up to +1.04‰) than its consecutive PM$_{2.5}$ N-sample.

The more positive $\Delta^{199}$Hg values measured for the PM$_{2.5}$ D-samples are highly unlikely uncharacteristic of known emission sources of PM$_{2.5}$-Hg. It is possible that PM$_{2.5}$-Hg from different emission sources may have different Hg isotope compositions. However, prior studies

showed that $\Delta^{199}$Hg values of the PBM from dominant anthropogenic emission sources are generally negative or close to zero. Schleicher et al. (2015) demonstrated that coal combustion is likely the major source of PM$_{2.5}$-Hg in Beijing. Huang et al (2016) reported that regional anthropogenic activities such as coal combustion ($\Delta^{199}$Hg values from −0.30‰ to 0.05‰), metal smelting (−0.20‰ to −0.05‰) and cement production (−0.25‰ to 0.05‰), as well as

biomass burning (low to −0.53‰) were likely the dominant sources of PM$_{2.5}$-Hg at this study site. As shown in Figure 2 and Table S1, the PM$_{2.5}$ D-samples with high $\Delta^{199}$Hg values (> 0.60‰) each had very different air mass back trajectories (Fig. S1). For instance, Sept-18-D (with $\Delta^{199}$Hg value +0.90‰), Sept-26-D (with $\Delta^{199}$Hg value +1.04‰) and Oct-3-D (with $\Delta^{199}$Hg value +0.86‰) were associated with north, southwest, and north-south mixed air

masses, respectively. A reasonable explanation of these observations is that high positive $\Delta^{199}$Hg values measured for D-samples resulted from PM$_{2.5}$-Hg transformation, specifically photoreduction, during atmospheric transport. Indeed, the diel variation of $\Delta^{199}$Hg for PM$_{2.5}$ D-N sample pairs may well reflect strong (D) versus less or no (N) influences of photochemical reactions under time-variant local and regional weather conditions.

**3.3   Photochemical reduction as a cause of odd-MIF in subset of daytime PM$_{2.5}$ samples**



To detail the effects of photochemical reactions on the variation of Hg isotope ratios for PM$_{2.5}$-Hg, we regrouped our dataset into subsets corresponding to day and night. We further regrouped our results into two source-related subsets, south-east (S-E) and north-west (N-W), according to the air mass backward trajectories during each sampling event (Fig. 1), and two other subsets

corresponding to sunny days within the S-E group (sunS-E) and all sunny days (Sun), which includes sunS-E and N-W as N-W consisted entirely of sunny days. The N-W subset of PM$_{2.5}$ was associated with an air mass that tracked from the north, northeast, northwest and west, relatively less polluted areas, and is therefore representative of long-range transport and relatively constant sources of PM$_{2.5}$ and Hg (Huang et al., 2016). The S-E subset was associated

with an air mass that tracked from the south, southwest, southeast and east, heavily polluted and highly populated areas, and was characterized by relatively high contents of PM$_{2.5}$, likely from industrial sources in the region (coal fired power plants, coking and steel industries). Unlike the N-W arriving air mass which corresponded to all sunny days during the entire sampling period, the S-E arriving air mass was associated with a range of weather conditions

including hazy, cloudy, rainy, and sunny days. According to our results (Table S2 and Fig. 2), PM$_{2.5}$ concentrations of the N-W subset ($23 \pm 19$ μg m$^{-3}$) were significantly ($p < 0.05$) lower than the S-E subset ($69 \pm 40$ μg m$^{-3}$), which is consistent with the fact that the N-W areas of Beijing were less industrialized, less populated and less polluted than the S-E areas. However, regardless of whether their associated air masses originated from moderately or heavily polluted

areas, both N-W and S-E subset samples showed diel variations in their Hg contents and isotope ratios (see Fig. 3 and discussion below). This, as discussed above, indicates that air mass source was not a dominant factor producing the diel variation of Hg isotope ratios in consecutive D-N PM$_{2.5}$ samples.

The observed diel difference in $\Delta^{199}$Hg values of PM$_{2.5}$-Hg is even more prominent and

statistically robust within subsets of PM$_{2.5}$ samples regrouped according to their air mass



trajectories (i.e., PM$_{2.5}$ source related) and sunny days (with greater extent of photochemical

reactions). As shown in Figure 3, $\Delta^{199}$Hg values for N-W subset samples collected during the

day had a higher range (0.04‰ to 0.90‰) and mean (0.39‰ ± 0.27‰ SD, $n$ = 10) compared

to those (−0.07‰ to 0.32‰, mean = 0.09‰ ± 0.13‰ SD, $n$ = 9) for N-samples ($p$ = 0.02).

Similarly, analysis of the sunS-E subset revealed a significant difference in $\Delta^{199}$Hg values ($p$ =

0.03) between sunny days and nights, but not when the entire S-E sample set ($p$ = 0.22), which

includes hazy, raining, and cloudy days, was considered. Since the N-W subset was associated

with less polluted areas and the S-E subset was associated with heavily polluted and highly

populated areas, the observation of significant diel variation of $\Delta^{199}$Hg in PM$_{2.5}$-Hg within each

subset (Fig. 3) is consistent with the above conclusion that such variation of PM$_{2.5}$-Hg isotope

ratios was not controlled by variation of Hg emission sources. The highly positive $\Delta^{199}$Hg

values observed for daytime samples within the Sun subset (Fig. S2) further supports the

conclusion that PM$_{2.5}$-Hg was strongly affected by photochemical reactions on sunny days.

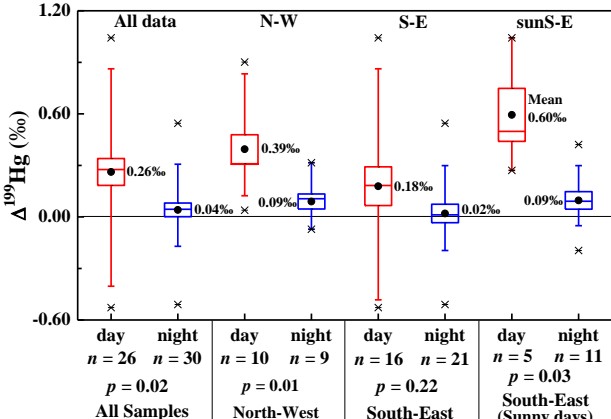

**Figure 3.** Diel variations of $\Delta^{199}$Hg for PM$_{2.5}$-Hg for the entire dataset, the N-W subset, the S-
E subset, and the S-E sunny days only subset. Note that all days included in the N-W subset
were sunny. Diel differences within each subset were examined using the Independent Samples
T-Test.


Linear correlations of $\Delta^{199}$Hg versus $\Delta^{201}$Hg for all 56 PM$_{2.5}$ samples (Fig. 4a) and three

subsets, N-W (Fig. 4b), S-E (Fig. 4c), and Sun (Fig. 4d), yielded slopes of $1.06 \pm 0.05$ (1SD, $r^2$

$= 0.89$), $1.06 \pm 0.12$ ($r^2 = 0.81$), $1.13 \pm 0.05$ ($r^2 = 0.92$) and $1.13 \pm 0.08$ ($r^2 = 0.84$), respectively

(Fig. 4). Such slopes are all indicative of photochemical reduction of Hg$^{2+}$ according to prior

studies (Bergquist and Blum, 2007; Zheng and Hintelmann, 2009). The photoreduction process

is further evidenced by a progressive increase in $\Delta^{199}$Hg from zero or slightly negative values

to positive values as the content of Hg in PM$_{2.5}$ ($C_{Hg}$) decreased in D-samples (Fig. S1a). This

trend is statistically more significant ($p < 0.05$) for D-samples within the N-W and Sun subsets

(Figs. S1-b and -d). Similarly, for all sunny day samples, a positive correlation ($p < 0.05$) was

also observed between $\Delta^{199}$Hg and $\delta^{202}$Hg (Fig. S3), consistent with prior experimental results

(Bergquist and Blum, 2007; Zheng and Hintelmann, 2009). Collectively, the Hg isotope results

suggest that photochemical reduction is an important process during the transport of PM$_{2.5}$-Hg

in the atmosphere.

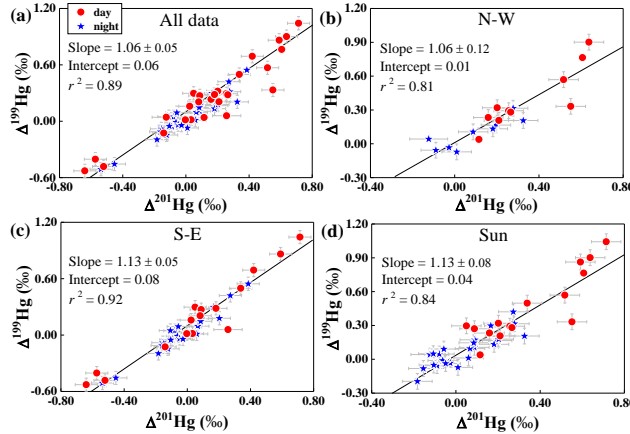

**Figure 4.** Correlations between $\Delta^{199}$Hg and $\Delta^{201}$Hg for different subsets of PM$_{2.5}$ samples: a)
all data, b) North-West (N-W), c) South-East (S-E) and d) All sunny days (Sun). The slope,
intercept and $r$-square of the line from simple linear regression. Vertical and horizontal error
bars correspond to 2SD analytical precision.





Among all D-samples, $\Delta^{199}$Hg is only weakly correlated with sunshine duration ($r^2 = 0.20$, $p$

= 0.02). However, $\Delta^{199}$Hg values for all D-samples collected on days with sunshine

durations >8 hrs are positive whereas half of the $\Delta^{199}$Hg values for samples collected on cloudy

or hazy days with shorter sunshine durations are negative or near zero (Fig. S4). In addition, a

significant positive linear correlation between $\Delta^{199}$Hg values and atmospheric ozone contents

($P_{O3}$) ($r^2 = 0.517$, $p < 0.01$) for all but four daytime samples was obtained (Fig. S5). The four

outliners (Sept-16-D, Sept-17-D, Oct-5-D and Oct-6-D) were collected on days with high ozone

($P_{O3}$ above 50 ppbv) and severe smog formation. Conversely, no significant correlation ($p >$

0.05) between $\Delta^{199}$Hg and $P_{O3}$ was found for the nighttime samples.

The increase in $\Delta^{199}$Hg of daytime PM$_{2.5}$-Hg with sunlight duration and ozone concentration

indicates that the physical and photochemical conditions of the atmosphere may affect the

atmospheric transformation of PM$_{2.5}$-Hg. A prior experimental study showed that GEM

oxidation can produce negative $\Delta^{199}$Hg values in oxidized Hg$^{2+}$ with $\Delta^{199}$Hg/$\Delta^{201}$Hg ratios of

1.6 and 1.9 for Br and Cl radical initiated oxidations (Sun et al., 2016). We can exclude the

possible contribution of Hg$^0$ oxidation to PM$_{2.5}$-Hg, given the fact that $\Delta^{199}$Hg/$\Delta^{201}$Hg ratio was

about 1.1 and most PM$_{2.5}$-Hg samples collected during the daytime when Br and Cl radicals

could form had positive $\Delta^{199}$Hg values. Thus it is highly unlikely that oxidation would have

caused the diel variation in Hg isotopes in PM$_{2.5}$. However, the exception to the observed

relationships between $\Delta^{199}$Hg with sunlight duration, and ozone concentration show that in a

highly oxidizing atmosphere (higher $P_{O3}$) such as occurs during extreme smog events, the odd-

MIF of Hg isotopes in PM$_{2.5}$ may decrease or reverse. A possible explanation for this effect

may be the increased production of GOM and its collection with PM$_{2.5}$-Hg during such smog

events. While PM$_{2.5}$-Hg samples collected on quartz fiber filters may include some GOM

(Lynam and Keeler, 2002), this contribution was likely small in most of our D and N samples

due to the opposing diel trends in the concentrations of PBM and GOM in urban air (Engle et





al., 2010). GOM would therefore not have had a major effect on the observed diel variations of

$\Delta^{199}$Hg values for PM$_{2.5}$-Hg and may have in fact masked an even larger MIF signature due to

the photoreduction of PBM during the day.

Interestingly, negative $\Delta^{199}$Hg values in daytime PM$_{2.5}$-Hg were only observed during a rainy

day and an extreme smog event. Scavenging of locally produced GOM during rain or smog

events may therefore have contributed to the reversal of the odd-MIF signature of Hg collected

as PM$_{2.5}$ at these times. In addition, the negative $\Delta^{199}$Hg values in PM$_{2.5}$ may have resulted from

the contribution of biomass burning with limited photoreduction effect during period of less

sunshine (Fig. 2 and Table S1) since plant foliage has negative $\Delta^{199}$Hg values (Yu et al., 2016)

and more negative $\Delta^{199}$Hg values (down to −0.53‰) of PM$_{2.5}$-Hg in Beijing were related to

biomass burning, a source of PM$_{2.5}$-Hg south of Beijing in autumn (Huang et al., 2016). This

could further explain the relatively lower $\Delta^{199}$Hg values in the majority of the N-samples (for

example, Sept-28-N and Oct-5-N with $\Delta^{199}$Hg of −0.46‰ and −0.51‰), even in those collected

under clear weathering condition. Indeed, each bulk sample collected during night time was a

mixture of the leftover PM$_{2.5}$ (with positive odd-MIF) from the previous daytime and the PM$_{2.5}$

newly input from various sources including industrial emissions (with close to zero $\Delta^{199}$Hg)

and biomass burning (somewhat negative $\Delta^{199}$Hg) (Huang et al., 2016) during nighttime.

While our results cannot exclude the effects of other possible processes, such as oxidation,

adsorption (and desorption), and precipitation, based on the limited previous studies (Jiskra et

al., 2012; Smith et al., 2015; Sun et al., 2016), these processes are not likely to be important to

the diel variation of odd-MIF of Hg isotopes in PM$_{2.5}$-Hg we observed.


### 3.4    Photochemical reduction as a cause of diel variation in odd-MIF in day-night sample pairs of PM$_{2.5}$





To explore the possible causes of diel variation in odd-MIF of Beijing PM$_{2.5}$-Hg further, we examined four subgroups of PM$_{2.5}$ samples, each of which included 2 to 4 consecutive pairs of

D-N samples that were collected during time periods of relatively constant atmospheric conditions (i.e., not being hazy, rainy or windy or having extremely high ozone ($P_{O3}$ above 50 ppbv)). Within each of the four subgroups, $\Delta^{199}$Hg and $\delta^{202}$Hg values were lower at night than during the previous or following day (Figs. 5-a and -b). As shown in Figure 5c, there is a significant positive correlation ($p < 0.01$) between $\Delta^{199}$Hg and $\delta^{202}$Hg values for all samples in

these four subgroups, with average values for both day and night falling right on the best fit line. The slope of this line is 1.15 ± 0.33, which is consistent with the reported value of 1.15 ± 0.07 for photochemical reduction of Hg$^{2+}$ in aqueous solution (Bergquist and Blum, 2007). Coincidently, the contents of Hg in PM$_{2.5}$ ($C_{Hg}$) were higher in N-samples than in immediately preceding or following D-samples (Fig. 5d), indicating a negative linear relationship between

$\Delta^{199}$Hg values and $C_{Hg}$ (Fig. 5e). Moreover, eight of the total eleven daytime samples among the four subgroups showed a positive linear correlation between $\Delta^{199}$Hg and the total cumulative daily solar radiation on a horizontal surface (SH) (Fig. 5f). These eleven samples also showed a negative correlation between the logarithmic values of $C_{Hg}$ and SH (Fig. 5g). These correlations are consistent with the photochemical reduction of divalent Hg observed

under laboratory conditions, and thus strongly support the hypothesis that photochemical reduction is an important process controlling the fate of ambient atmospheric PM$_{2.5}$-Hg. Given its diel trend and relatively large range, the MIF of odd Hg isotopes in Beijing PM$_{2.5}$ we observed was most likely due to the magnetic isotope effect (MIE), which has been invoked to explain MIF during the photochemical reduction of aqueous Hg$^{2+}$ (Zheng and Hintelmann,

2010b).





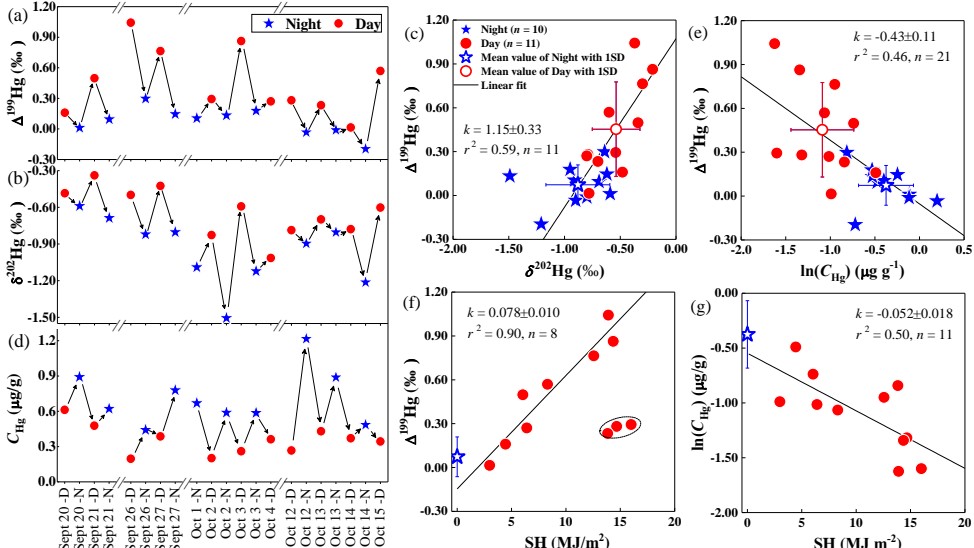

**Figure 5.** Hg isotope ratios and contents in four subgroups of consecutive pairs of day-night samples collected during periods of relatively constant atmospheric conditions. Linear correlations between $\Delta^{199}$Hg and $\delta^{202}$Hg (c), $C_{Hg}$ (e) and the total cumulative daily solar radiation on a horizontal surface (SH, MJ m$^{-2}$) (f), and between $C_{Hg}$ and SH (g) were displayed.

### 3.5 Even isotope MIF

Diel variation of $\Delta^{200}$Hg signatures was also observed in PM$_{2.5}$-Hg. Prior studies reported mainly positive $\Delta^{200}$Hg values for wet precipitation (up to 1.24‰) (Gratz et al., 2010; Chen et al., 2012; Wang et al., 2015; Yuan et al., 2015) and aerosols (−0.05‰ to 0.28‰) (Rolison et al., 2013; Das et al., 2016; Huang et al., 2016). Even Hg isotope MIF has been attributed to complex redox processes occurring in the upper atmosphere, but the underlying mechanisms remain unclear (Mead et al., 2013; Eiler et al., 2014). Thus, the even-MIF signatures may suggest a small proportion of PM$_{2.5}$-Hg is likely originated from the upper atmosphere, through for example long-term transport. Given the fact that all samples displayed only slightly positive $\Delta^{200}$Hg (average 0.07‰ ± 0.06‰, 1SD), this contribution may be very limited. In addition, $\Delta^{200}$Hg values are weakly, but significantly correlated with $\Delta^{199}$Hg ($r^2 = 0.13$, $p < 0.01$) (Fig.



S6) and $\delta^{202}$Hg ($r^2 = 0.27$, $p < 0.01$) (Fig. S7). No mechanistic explanation is available yet for such observations, however.


## 4 Conclusions

This study showed significant diel variations of Hg isotopic compositions for ambient PM$_{2.5}$-Hg collected in the city of Beijing. The Hg isotope signatures featured a large range of MDF ($\delta^{202}$Hg value from −1.49‰ to 0.55‰, mean of −0.53‰ ± 0.40‰) and significant ($p < 0.05$)

MIF with more positive $\Delta^{199}$Hg values in daytime samples (0.26‰ ± 0.40‰) than at night (0.04‰ ± 0.22‰). The results clearly indicated that the Hg isotope compositions of PM$_{2.5}$-Hg are impacted variously by both weather conditions (such as sunlight duration), which may promote the photochemical reaction, and directions of air mass trajectories, which are related to possible sources of PM$_{2.5}$. D-N paired samples having similar air mass backward trajectories and hence

similar sources exhibited strong positive correlations between $\Delta^{199}$Hg and $\Delta^{201}$Hg with a slope of 1.1 and $\Delta^{199}$Hg and $\delta^{202}$Hg with a slope of 1.15, and a decrease in the content of Hg in PM$_{2.5}$ as $\Delta^{199}$Hg increased. These results provide isotopic evidence that local, daytime photochemical reduction of divalent Hg is of critical importance to the fate of PM$_{2.5}$-Hg in urban atmosphere. Although the specific reactions and mechanisms that control Hg isotope fractionation (MDF

and MIF) in Beijing PM$_{2.5}$ could not be explicitly determined from this field study, our result illustrated that, in addition to variation in sources, photochemical reduction appears to be an important process that affects both the content and isotopic composition of PM$_{2.5}$-Hg. Further systematic study is thus needed to better quantify the photoreduction of PM$_{2.5}$-Hg to estimate the percentage of reduced Hg it produces and its impact on the global biogeochemical cycling

of Hg.





*Acknowledgments.* This study was supported financially by the National Key Research and
Development Program of China (No. 2017YFC0212702), Natural Science Foundation of China
(no. 41701268, 41625012, 41273023), Guizhou Scientific Research Program (No. 20161158)

and State Key Laboratory of Organic Geochemistry (OGL-201501).

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
