# Peer review of "Diel variation of mercury stable isotope ratios record photoreduction of PM2.5-bound mercury"

_Atmospheric Chemistry and Physics, 2018_

## Referee Comment (RC1) · Anonymous Referee #1 · 11 Oct 2018

Huang et al. present a study on diurnal particulate (aerosol) Hg isotope variation in Beijing. I have reviewed this MS previously for ES&T and I was curious to see how the MS evolved following my previous suggestions. I regret to say that these have been largely ignored, so I paste here my previous review because it is still of interest:

"Huang et al.'s study is unique and the variations observed, with lower Hg concentrations and higher Hg MIF during daytime, are rather interesting and novel. The authors interpret this as evidence for in-aerosol photoreduction of Hg, which would be an important result if it were true. The MS is fairly well written, organized and cited. Although at first sight I agree with the interpretation, I find that the authors privilege the photoreduction interpretation without much in-depth discussion of alternative interpretations. For example, the current gas-aerosol partitioning model (Amos et al., 2012, ACP) sug-

gests divalent gaseous Hg (GOM) to partition to aerosol at low temperatures. The authors should estimate and discuss if this process can lead to their concentration observations. I suggest the authors consider the influence of boundary layer dynamics and stratification: daytime turbulence could lead to mixing of above lying, cleaner free tropospheric air with high MIF, whereas nighttime stratification traps Hg emissions with low MIF."

In the current MS submitted to ACP I did not find discussion of gaseous Hg(0) – aerosol partitioning, nor discussion of boundary layer dynamics. The authors should discuss with atmospheric physicists, and see if proxies of boundary layer mixing can be used. For ex. PM2.5 itself seems higher during nighttime than daytime, which is likely due to nighttime boundary layer stratification which traps pollutant emissions. Daytime heating of land and ensuing turbulence will mix boundary layer air with overlying free tropospheric air. Such mixing may, or may not, generate all the trends observed. It should be discussed and counter-argumented.

In summary, I am convinced that the dataset is novel and of strong interest to the atmospheric Hg community and ACP readers, but I suggest the authors to better think through alternative interpretations, and to respect the reviewing process. The editor and reviewers spend time to try and make your study better.

Minor comments: L99. "It is intuitive that, while both D and N PM2.5 samples may have similar local or regional sources if the wind trajectory remains unchanged, D samples could have been exposed to more solar radiation than N samples, likely resulting in diel variations in the Hg isotope compositions that are indicative of differences in photochemical transformation of PM2.5-Hg."

Not sure this makes sense: if PM-Hg was emitted 1 week ago and travelled across China, the particles went through 7 day/night periods, all receiving more or less the same amount of radiation.

L211. The statistics of diel variations are discussed here, with reference to Table S3. It

appears to me that paired T-tests should be reported in the text, and not means and p values for the whole dataset. A key question is whether PM2.5 shows diel variation in the paried T-test? The p values in the text do not correspond to metrics in Table S3, so the discussion is hard to follow.

L171. Why 24h backtrajectories and not more? What is known about PBM lifetime in the Chinese boundary layer? I think there is a discussion in Horrowitz et al., ACP, 2017 on this.

L342. "Interestingly, negative D199Hg values in daytime PM2.5-Hg were only observed during a rainy day and an extreme smog event. Scavenging of locally produced GOM during rain or smog events may therefore have contributed to the reversal of the odd-MIF signature of Hg collected as PM2.5 at these times."

Why would local GOM have a negative D199? Not clear to reader.

---

## Referee Comment (RC2) · Anonymous Referee #2 · 19 Oct 2018

This manuscript quantified the diel variation of Hg isotope compostion of particulate-bound mercury (PBM) and revealed that daily photochemical reduction of divalent Hg is of critical importance to the fate of PM2.5-Hg in urban atmospheres. The topic is quite interesting and is important for understanding global mercury cycling. Publication is suggested after minor revision.

Line 114 Is one air sampler enough? PBM concentration in the air is quite small. To obtain enough mercury for isotope analysis, especially when the sampling time was reduced, it seems that we need more samplers.

Line 163 Why do you choose the height of 500 m?

Figure 2 This figure is too busy. Instead listing all data according to time series, is is

possible to classify the figure into several subgroup according to the topic you wants to discussed? This figure can be moved to supporting information.

Figure 2(a) How to explain the negative value of $\Delta$199Hg on Sep 28?

Figure 2(f) all legends are suggested to be listed on the top of this figure. It is diffult to find "clear" "cloudy" "rain" in this figure.

Line 356 "While our results cannot exclude the effects of other possible processes, such as oxidation, adsorption (and desorption), and precipitation, based on the limited previous studies (Jiskra et al., 2012; Smith et al., 2015; Sun et al., 2016), these processes are not likely to be important to the diel variation of odd-MIF of Hg isotopes in PM2.5-Hg we observed." The observed isotope factionation is a phenomenon while the photochemical reduction is one process leading to this phenomenon. How can you exclude the impact from other processes? Evidences are required to prove this conclusion.

Figure 5(a) What is the main reason that caused the variation of $\Delta$199Hg during the night time? Is it possible caused by measurement error? If this is ture, it is better to point out this in method part.

---

## Author Comment (AC1) · 8 Nov 2018

We thank the reviewers for their constructive comments and suggestions. We have completed the revision of the manuscript according to the comments and suggestions provided by the reviewers. We appreciate very much all comments made by the reviewers; they are very valuable for improving the readability of the manuscript and interpretation of the protocol. Detailed Response, Marked Manuscript and Revised Supporting Information were also combined into a PDF file in supplement. In the marked version, the revised areas are in red color. Please check the file in supplement. Blow we have compiled our point-by-point responses to the comments.

[Figure]

**Detailed Responses to Referees**

**1 Anonymous Referee 1**

**1.1** Huang et al. present a study on diurnal particulate (aerosol) Hg isotope variation in Beijing. I have reviewed this MS previously for EST and I was curious to see how the MS evolved following my previous suggestions. I regret to say that these have been largely ignored, so I paste here my previous review because it is still of interest: "Huang et al.'s study is unique and the variations observed, with lower Hg concentrations and higher Hg MIF during daytime, are rather interesting and novel. The authors interpret this as evidence for in-aerosol photoreduction of Hg, which would be an important result if it were true. The MS is fairly well written, organized and cited.

– Thank you for your comments. We have addressed the comments below.

**1.2** Although at first sight I agree with the interpretation, I find that the authors privilege the photoreduction interpretation without much in-depth discussion of alternative interpretations. For example, the current gas-aerosol partitioning model (Amos et al., 2012, ACP) suggests divalent gaseous Hg (GOM) to partition to aerosol at low temperatures. The authors should estimate and discuss if this process can lead to their concentration observations.

– Thanks for the above comments. In this study, we observed that PBM (mass-based concentration) is greater during night-time than daytime for consecutive day-night pairs. We agree with the reviewer that the PBM is likely affected by gas-aerosol partitioning of GOM, in addition to the impact of sources. It is intuitive that the partitioning equilibrium depends on both temperature and GOM level. Adsorption or partitioning of GOM from air to PM is an exothermic process; lower temperature at night-time favors stronger adsorption of GOM on PM than at day-time. For example, Rutter and Schauer (2007) and Amos et al. (2012) proposed gas-aerosol partitioning models, which suggest that more divalent gaseous Hg (GOM) may partition onto aerosols at lower temperature. If we assume that GOM could remain a constant level during daytime and night-time,

relatively lower temperature at night-time should result in higher PBM in the night-time than in the daytime. However, the assumption of constant GOM over day-night time is likely untrue. According to several recent studies, the GOM measured in the field also exhibits significant diel variation, with higher GOM concentrations found during the daytime than at night (Lan et al., 2012; Liu et al., 2007; Poissant et al., 2005), likely due to the photo-oxidation of GEM. For example, the GOM measured during the spring season in Salt Lake city was equal to or lower than 4 pg m$^{-3}$ at night-time and was as high as 20 pg m$^{-3}$ during daytime (Lan et al., 2012). Another study reported that the GOM measured in the urban area of Detroit, Michigan was lower than 9 pg m$^{-3}$ at night and as high as 13 pg m$^{-3}$ during day-time (Liu et al., 2007). Apparently, the temperature effect on PBM concentration due to favored adsorption of GOM during night-time (lower temperature) may be partially (if not totally) off-set by lower GOM levels during night-time. In other words, the net PBM may show less diel variation as predicted from the temperature-dependent partitioning model when GOM is also substantially lowered during the night-time.

To support the above argument, we used an inverse approach and a GOM partitioning model to compute hypothetic GOM levels corresponding to each of our PBM observations at ambient temperature. We used the GOM gas-aerosol partitioning model proposed by Amos et al., 2012, which has the following equation: $\log 10(K^{-1}) = (10\pm1)$ – $(2500\pm300)/T$, where $K = (PBM/PM_{2.5})/GOM$ with PBM and GOM in common volumetric units (pg m$^{-3}$), PM$_{2.5}$ in $\mu$g m$^{-3}$, and T in K. We used the measured PM$_{2.5}$-Hg as PBM and assumed that the PM$_{2.5}$-Hg measured for each sample is 100

Similarly, gas-aerosol partitioning of GOM does not likely account for the diel variation of PM$_{2.5}$-Hg concentrations measured in this study. Meanwhile, our data showed that the average $\Delta^{199}$Hg value during the daytime (0.26‰ $\pm$ 0.40‰ 1SD, n = 26) is (statistically) significantly (p < 0.05, t-test) higher than during the nighttime (0.04‰ $\pm$ 0.22‰ 1SD, n = 30). This slight diel variation of odd-MIF of Hg isotopes was explained in terms of photoreduction of PBM during daytime. In addition we argue that the diel variations of

odd-MIF of Hg isotopes does not result from GOM gas-aerosol partitioning. In general, divalent Hg gas-aerosol partitioning is considered as chemisorption and desorption (Rutter and Schauer, 2007). Prior studies showed that the adsorption/desorption and precipitation of aqueous $Hg^{2+}$ had insignificant odd-MIF of Hg isotopes (Jiskra et al., 2012; Smith et al., 2015), suggesting that the GOM partitioning process may not result in the characteristics of odd-MIF of Hg isotopes we observed for the $PM_{2.5}$ samples.

In conclusion, gas-particle partitioning may increase PBM during the night-time due to relatively lower temperature compared to the daytime. The actual increase of PBM during the nighttime may be off-set by lower GOM levels during nighttime when little or no production of GOM by the photo-oxidation of GEM may occur. It is shown that GOM is high during daytime likely due to stronger photo-oxidation. GOM can be adsorbed on to PM where the active Hg species are also photo-reduced to elemental mercury. Such a dynamic and complex adsorption-photoreduction cycle yields lowered PBM levels, along with characteristic Hg isotope properties. In other words, our thesis is that the photochemical reactions cause the concentration reduction of $PM_{2.5}$-Hg as well as fractionation of the $PM_{2.5}$-bound Hg isotope compositions.

To address these issues, we have included a discussion about the possible effects of gas-aerosol partitioning on the diel variation of PBM. See the revised manuscript on line 358 to 370, it reads: "A possible explanation of the observed effects of diel variation of $PM_{2.5}$-Hg would be the temperature-dependent gas-aerosol partitioning of GOM (Amos et al., 2012; Rutter and Schauer, 2007), which favors more adsorption of GOM on PM during nighttime when atmospheric temperature us relatively lower than daytime. However, the magnitude of such adsorption is also proportional to the GOM concentration in the atmosphere. An inverse calculation exercise (in SI) shows that the higher $PM_{2.5}$-Hg measured for our samples would require higher GOM concentrations during the nighttime, which contradicts with prior findings that GOM concentrations are significantly lower during the nighttime than the daytime as GOM is a product of photo-oxidation processes (Amos et al., 2012; Liu et al., 2007; Poissant et al., 2005). In

addition, GOM gas-aerosol partitioning is considered a chemisorption and desorption process (Rutter and Schauer, 2007), which unlikely result in appreciable odd-MIF of Hg isotopes (Jiskra et al., 2012; Smith et al., 2015). Therefore, GOM partitioning would have little or no effect on the observed diel variations of $\Delta^{199}$Hg values for PM$_{2.5}$-Hg."

**1.3** I suggest the authors consider the influence of boundary layer dynamics and stratification: daytime turbulence could lead to mixing of above lying, cleaner free tropospheric air with high MIF, whereas nighttime stratification traps Hg emissions with low MIF."

– Thank you for your suggestion. We agree that the boundary layer was higher during the daytime than at night, and daytime turbulence could help to mix the air between bottom and top of the layer. As your suggest, with constant Hg emission and PM$_{2.5}$ deposition rates, Hg$^{2+}$ photoreduction on PM$_{2.5}$ during the daytime may be enhanced at the top of the boundary layer (up to 1500 m) on a sunny day and produce much more positive odd-MIF of Hg isotopes on PM$_{2.5}$, while at night, the lower boundary layer traps a portion of daytime PBM at much low altitudes (mean of about 300 m). The mixing of residual daytime PBM with newly emitted PBM in the thinner boundary layer at night may help to explain why nighttime PBM had odd-MIF values closer to source emissions.

Per your suggestions and comments, we have added a discussion about the possible effects of the difference of boundary layer thickness during daytime and nighttime on the diel variations of Hg isotope ratios in the PM$_{2.5}$ samples we collected. See the revised manuscript on line 371 to 381, it reads: "Variation in atmosphere boundary layer height (ABLH) from 1000 to 1300 m during daytime to less than 200 to 300 m during nighttime may have contributed to the diel variation in Hg isotopic composition of PM$_{2.5}$-Hg (Quan et al., 2013). With a high ABLH during daytime, relatively strong turbulence may help mixing the PM$_{2.5}$-Hg from the surface to the upper free troposphere, where photoreactions may be favored due to higher intensities of ultraviolet radiation on clear days. In contrast, a lower ABLH at night may weaken the vertical transport of PM$_{2.5}$-

Hg, but enhance the contribution from newly produced $PM_{2.5}$-Hg, possibly resulting in higher concentrations of $PM_{2.5}$-Hg with negative or close to zero $\Delta^{199}$Hg values from emission sources and/or GOM. However, vertically-resolved, day-night measurements of Hg stable isotope ratios in PBM and GOM are needed to fully evaluate the effects of various physical processes on diel variation of the Hg isotopic compositions for the $PM_{2.5}$".

**1.4** In the current MS submitted to ACP I did not find discussion of gaseous Hg(0) – aerosol partitioning, nor discussion of boundary layer dynamics. The authors should discuss with atmospheric physicists, and see if proxies of boundary layer mixing can be used. For ex. $PM_{2.5}$ itself seems higher during nighttime than daytime, which is likely due to nighttime boundary layer stratification which traps pollutant emissions. Daytime heating of land and ensuing turbulence will mix boundary layer air with overlying free tropospheric air. Such mixing may, or may not, generate all the trends observed. It should be discussed and counter-argumented.

– We agree that, although $PM_{2.5}$ concentrations had insignificant diel variation (p = 0.887, paired samples t-test), the change of boundary layer thickness between daytime and nighttime could affect PBM transformations, and we now address the possibility of this effect as described in our comments above.

**1.5** In summary, I am convinced that the dataset is novel and of strong interest to the atmospheric Hg community and ACP readers, but I suggest the authors to better think through alternative interpretations, and to respect the reviewing process. The editor and reviewers spend time to try and make your study better.

– Thank you for your suggestions and comments. Although the editor of EST did not give us a chance to response your comments, we truly appreciate the editor and reviewers for their comments on this manuscript. We are very glad to have this chance to respond to your comments at here, and we revised the manuscript accordingly.

**1.6** Minor comments:

L99. "It is intuitive that, while both D and N PM$_{2.5}$ samples may have similar local or regional sources if the wind trajectory remains unchanged, D samples could have been exposed to more solar radiation than N samples, likely resulting in diel variations in the Hg isotope compositions that are indicative of differences in photochemical transformation of PM$_{2.5}$-Hg." Not sure this makes sense: if PM-Hg was emitted 1 week ago and travelled across China, the particles went through 7 day/night periods, all receiving more or less the same amount of radiation.

– We have deleted this sentence.

L211. The statistics of diel variations are discussed here, with reference to Table S3. It appears to me that paired T-tests should be reported in the text, and not means and p values for the whole dataset. A key question is whether PM$_{2.5}$ shows diel variation in the paired T-test? The p values in the text do not correspond to metrics in Table S3, so the discussion is hard to follow.

– We have revised this paragraph per your suggestions, which now reads (on line 207 to 215 in the revised manuscript): "T-test results (Table S3) showed that diel variation was statistically significant (p < 0.05) for Hg contents, $\Delta^{199}$Hg, and $\Delta^{200}$Hg values, as their p values are 0.005, 0.000 and 0.004 resulting from paired samples t-test, and are 0.003, 0.017 and 0.019 resulting from independent samples t-test. For all samples, Hg contents for D-samples (0.32 $\pm$ 0.14 $\mu$g g-1) were lower than N-samples (0.48 $\pm$ 0.24 $\mu$g g-1), and $\Delta^{199}$Hg and $\Delta^{200}$Hg values for D-samples (mean of 0.26‰ $\pm$0.40‰ and 0.09‰ $\pm$ 0.06‰ respectively) were higher than N-samples ($-$0.04‰ $\pm$ 0.22‰ and 0.06‰ $\pm$ 0.05‰ respectively). However, PM$_{2.5}$ concentrations and $\delta$202Hg had statistically insignificant (p > 0.05) diel variation, as their p values are 0.887 and 0.052 resulting from paired samples t-test, and are 0.909 and 0.053 resulting from independent samples t-test".

L171. Why 24h back trajectories and not more? What is known about PBM lifetime in the Chinese boundary layer? I think there is a discussion in Horowitz et al., ACP, 2017

on this.

– Per your suggestion, we changed to 72-h back trajectories in the revised manuscript. The results of such 72-h back trajectory frequencies are shown that the dominant air masses (over 90

L342. "Interestingly, negative $\Delta^{199}$Hg values in daytime $PM_{2.5}$-Hg were only observed during a rainy day and an extreme smog event. Scavenging of locally produced GOM during rain or smog events may therefore have contributed to the reversal of the odd-MIF signature of Hg collected as $PM_{2.5}$ at these times." Why would local GOM have a negative $\Delta^{199}$Hg? Not clear to reader.

– We have revised these sentences for clarity (see the revised manuscript on line 341 to 346): "Interestingly, negative $\Delta^{199}$Hg values in daytime $PM_{2.5}$-Hg were only observed during a rainy day and an extreme smog event. Since the Hg emitted from local sources had close to zero and negative values of odd-MIF, higher humidity (such as during rainy days) and heavy pollution (the extreme smog) may enhance the effect of scavenging of locally produced gaseous or particulate Hg during rain or smog events, which may therefore have contributed to the reversal of the odd-MIF signature of Hg collected as $PM_{2.5}$ at these times".  

**2 Anonymous Referee 2**

**2.1** This manuscript quantified the diel variation of Hg isotope composition of particulate bound mercury (PBM) and revealed that daily photochemical reduction of divalent Hg is of critical importance to the fate of $PM_{2.5}$-Hg in urban atmospheres. The topic is quite interesting and is important for understanding global mercury cycling. Publication is suggested after minor revision.

– Thank you for your comments.

**2.2** Line 114 Is one air sampler enough? PBM concentration in the air is quite small. To obtain enough mercury for isotope analysis, especially when the sampling time was

reduced, it seems that we need more samplers.

– We used one sampler for collecting the PM samples. Among the 61 $PM_{2.5}$ samples we collected, 56 had sufficient Hg mass for Hg isotope analysis. It would be better if two or more samplers were used simultaneously for sampling so that (1) sufficient mass of PBM can be obtained for isotope analysis and (2) replicates could be used for isotope analysis.

**2.3** Line 163 Why do you choose the height of 500 m?

– In our back trajectory modeling, we used 500 m as the average boundary layer height in Beijing according to a prior study by (Xiang et al., 2019). The estimated backward HYSPLIT trajectories of air masses should be acceptable. Alternatively, we also used different arrival heights (200, 500, 1000 m above ground level) for estimating the backward trajectories. The results indicate that the transport pathways are not very sensitive to the selected heights within the studied area. Per your comments, we have added detail information in the revised version of the Supporting Information.

**2.4** Figure 2 This figure is too busy. Instead listing all data according to time series, is it possible to classify the figure into several subgroup according to the topic you wants to discussed? This figure can be moved to supporting information.

– Per your suggestion, we have revised Figure 2 as following, which shows the chronological sequence of MIF ($\Delta^{199}Hg$ and $\Delta^{200}Hg$) and MDF ($\delta^{202}Hg$) of the 56 samples collected during the daytime (D, red) and nighttime (N, blue), along with selected weather data including cumulative hours of sunshine (Solar) and air mass backward-trajectory directions.

**2.5** Figure 2(a) How to explain the negative value of $\Delta^{199}Hg$ on Sep 28?

– The explanation had been described in Line 363 to 373. "Interestingly, negative $\Delta^{199}Hg$ values in daytime $PM_{2.5}$-Hg were only observed during a rainy day and an extreme smog event. Since the Hg emitted from local sources had close to zero and

negative values of odd-MIF, higher humidity (such as during rainy days) and heavy pollution (the extreme smog) may enhance the effect of scavenging of locally produced gaseous or particulate Hg during rain or smog events, which may therefore have contributed to the reversal of the odd-MIF signature of Hg collected as PM$_{2.5}$ at these times. In addition, the negative $\Delta^{199}$Hg values in PM$_{2.5}$ may have resulted from the contribution of biomass burning with limited photoreduction effect during periods of less sunshine (Fig. 2 and Table S1) since plant foliage has negative $\Delta^{199}$Hg values (Yu et al., 2016) and more negative $\Delta^{199}$Hg values (down to $-0.53‰$ of PM$_{2.5}$-Hg in Beijing were related to biomass burning, a source of PM$_{2.5}$-Hg south of Beijing in autumn (Huang et al., 2016)."

**2.6** Figure 2(f) all legends are suggested to be listed on the top of this figure. It is difficult to find "clear" "cloudy" "rain" in this figure.

– We have revised the legends in the revised Figure 2(d).

**2.7** Line 356 "While our results cannot exclude the effects of other possible processes, such as oxidation, adsorption (and desorption), and precipitation, based on the limited previous studies (Jiskra et al., 2012; Smith et al., 2015; Sun et al., 2016), these processes are not likely to be important to the diel variation of odd-MIF of Hg isotopes in PM$_{2.5}$-Hg we observed." The observed isotope fractionation is a phenomenon while the photochemical reduction is one process leading to this phenomenon. How can you exclude the impact from other processes? Evidences are required to prove this conclusion.

– We agree that these observations may be the result of multiple processes. We now address the possible contributions of two additional physical process, adsorption of GOM to PM and reduced boundary layer mixing of PBM at night (see responses to Referee 1).

**2.8** Figure 5(a) What is the main reason that caused the variation of $\Delta^{199}$Hg during the night time? Is it possible caused by measurement error? If this is true, it is better to

point out this in method part.

– Our data showed that the $\Delta^{199}$Hg values ranged from 0.01‰ to 0.30‰ for the nighttime samples in Figure 5(a) and ranged from -0.51‰ to 0.55‰ for all nighttime samples in this study. The method we used for quantifying Hg isotopes bears an uncertainty (2SD) of 0.06‰ for $\Delta^{199}$Hg for the samples. Statistically, differences between $\Delta$199Hg values for daytime and nighttime samples were clearly significant and should not have been caused by uncertainty of the method. To help readers understand this issue, we have added the measurement uncertainty in the caption of Figure 5.

*Corresponding author.

E-mail: huangqiang@vip.gyig.ac.cn

We thank the reviewers for their constructive comments and suggestions. We have completed the revision of the manuscript according to the comments and suggestions provided by the reviewers. We appreciate very much all comments made by the reviewers; they are very valuable for improving the readability of the manuscript and interpretation of the protocol. Detailed Response, Marked Manuscript and Revised Supporting Information were also combined into a PDF file in supplement. In the marked version, the revised areas are in red color. Please check the file in supplement. Blow we have compiled our point-by-point responses to the comments.

**Detailed Responses to Referees**

**1 Anonymous Referee #1**

Huang et al. present a study on diurnal particulate (aerosol) Hg isotope variation in Beijing. I have reviewed this MS previously for ES&T and I was curious to see how the MS evolved following my previous suggestions. I regret to say that these have been largely ignored, so I paste here my previous review because it is still of interest: "Huang et al.'s study is unique and the variations observed, with lower Hg concentrations and higher Hg MIF during daytime, are rather interesting and novel. The authors interpret this as evidence for in-aerosol photoreduction of Hg, which would be an important result if it were true. The MS is fairly well written, organized and cited.

--Thank you for your comments. We have addressed the comments below.

Although at first sight I agree with the interpretation, I find that the authors privilege the photoreduction interpretation without much in-depth discussion of alternative interpretations. For example, the current gas-aerosol partitioning model (Amos et al., 2012, ACP) suggests divalent gaseous Hg (GOM) to partition to aerosol at low temperatures. The authors should estimate and discuss if this process can lead to their concentration observations.

--Thanks for the above comments. In this study, we observed that PBM (mass-based concentration) is greater during night-time than daytime for consecutive day-night pairs. We agree with the reviewer that the PBM is likely affected by gas-aerosol partitioning of GOM, in addition to the impact of sources. It is intuitive that the partitioning equilibrium depends on both temperature and GOM level. Adsorption or partitioning of GOM from air to PM is an exothermic process; lower temperature at night-time favors stronger adsorption of GOM on PM than at day-time.

For example, Rutter and Schauer (2007) and Amos et al. (2012) proposed gas-aerosol partitioning models, which suggest that more divalent gaseous Hg (GOM) may partition onto aerosols at lower temperature. If we assume that GOM could remain a constant level during daytime and night-time, relatively lower temperature at night-time should result in higher PBM in the night-time than in the daytime. However, the assumption of constant GOM over day-night time is likely untrue. According to several recent studies, the GOM measured in the field also exhibits significant diel variation, with higher GOM concentrations found during the daytime than at night (Poissant et al., 2005; Liu et al., 2007; Lan et al., 2012), likely due to the photo-oxidation of GEM. For example, the GOM measured during the spring season in Salt Lake city was equal to or lower than 4 pg m$^{-3}$ at night-time and was as high as 20 pg m$^{-3}$ during daytime (Lan et al., 2012). Another study reported that the GOM measured in the urban area of Detroit, Michigan was lower than 9 pg m$^{-3}$ at night and as high as 13 pg m$^{-3}$ during day-time (Liu et al., 2007). Apparently, the temperature effect on PBM concentration due to favored adsorption of GOM during night-time (lower temperature) may be partially (if not totally) off-set by lower GOM levels during night-time. In other words, the net PBM may show less diel variation as predicted from the temperature-dependent partitioning model when GOM is also substantially lowered during the night-time.

To support the above argument, we used an inverse approach and a GOM partitioning model to compute hypothetic GOM levels corresponding to each of our PBM observations at ambient temperature. We used the GOM gas-aerosol partitioning model proposed by Amos et al., 2012, which has the following equation: $\log_{10}(K^{-1}) = (10\pm1) - (2500\pm300)/T$, where $K = $ (PBM/PM$_{2.5}$)/GOM with PBM and GOM in common volumetric units (pg m$^{-3}$), PM$_{2.5}$ in $\mu$g m$^{-3}$, and $T$ in K. We used the measured PM$_{2.5}$-Hg as PBM and assumed that the PM$_{2.5}$-Hg measured for each sample is 100% in divalent and active mercury forms. The calculated GOM concentrations are presented in the following Table R1. In summary, the calculated GOM

concentrations range from 1.5 to 31 pg m$^{-3}$, with average values of 11±5 pg m$^{-3}$ during the daytime and 13±7 pg m$^{-3}$ during the nighttime. Overall, the calculated GOM exhibit insignificant ($p$ = 0195, paired samples $t$-test) diel variation of GOM concentration, i.e., there would be little or no difference of GOM between day- and night-time. Close inspection of the data (Table R1) showed that half of the paired day-night samples have higher calculated GOM concentrations during the nighttime than in daytime. This is opposite to the prior findings cited above (Poissant et al., 2005; Liu et al., 2007; Lan et al., 2012) showing higher measured GOM in the daytime than in the nighttime and indicates that processes other than gas-aerosol partitioning control GOM concentrations in the environment.

Similarly, gas-aerosol partitioning of GOM does not likely account for the diel variation of PM$_{2.5}$-Hg concentrations measured in this study.

Table R1. Calculated GOM concentrations of day and night samples. The value of GOM concentrations higher at night than the consecutively days are in bold text.

| Daytime samples | Ave. T (°C) | Hg Con. ($\mu g\,g^{-1}$) | $K$ m$^3\,\mu g^{-1}$ | GOM pg m$^{-3}$ | Nighttime samples | Ave. T (°C) | Hg Con. ($\mu g\,g^{-1}$) | $K$ m$^3\,\mu g^{-1}$ | GOM pg m$^{-3}$ |
|---|---|---|---|---|---|---|---|---|---|
| | | | | | Sept-15-N | 19.8 | 0.52 | 0.035 | 15 |
| Sept-16-D | 24.1 | 0.41 | 0.026 | 16 | Sept-16-N | 21.1 | 0.44 | 0.032 | 14 |
| Sept-17-D | 24.8 | 0.38 | 0.025 | 15 | Sept-17-N | 22.6 | 0.47 | 0.029 | **16** |
| Sept-18-D | 27.3 | 0.17 | 0.021 | 8.0 | Sept-18-N | 21.7 | 0.31 | 0.030 | **10** |
| Sept-19-D | 26.1 | 0.09 | 0.023 | 3.9 | Sept-19-N | 21.7 | 0.29 | 0.030 | 10 |
| Sept-20-D | 24.5 | 0.61 | 0.025 | 24 | Sept-20-N | 21.9 | 0.89 | 0.030 | **30** |
| Sept-21-D | 25.2 | 0.48 | 0.024 | 20 | Sept-21-N | 22.6 | 0.62 | 0.029 | **22** |
| Sept-22-D | 22.9 | 0.53 | 0.028 | 19 | Sept-22-N | 18.3 | 0.31 | 0.038 | 8.1 |
| Sept-23-D | 23.9 | 0.15 | 0.026 | 5.7 | Sept-23-N | 21.6 | 0.54 | 0.031 | **18** |
| Sept-24-D | 23 | 0.38 | 0.028 | 14 | Sept-24-N | 17.7 | 0.2 | 0.040 | 5.0 |
| Sept-25-D | 24.4 | 0.38 | 0.025 | 15 | Sept-25-N | 17.3 | 0.14 | 0.041 | 3.4 |
| Sept-26-D | 23.7 | 0.2 | 0.027 | 7.5 | Sept-26-N | 20.5 | 0.44 | 0.033 | 13 |
| Sept-27-D | 23.8 | 0.39 | 0.026 | 15 | Sept-27-N | - | 0.78 | | |
| Sept-28-D | 18.1 | 0.32 | 0.039 | 8 | Sept-28-N | 17.4 | 0.34 | 0.041 | 8.4 |
| Sept-29-D | 15.7 | 0.48 | 0.046 | 11 | Sept-29-N | 14.7 | 0.36 | 0.049 | 7.4 |
| Sept-30-D | 18.1 | 0.16 | 0.039 | 4.1 | Sept-30-N | 15.5 | 0.64 | 0.046 | **14** |
| Oct-1-D | 19.4 | 0.12 | 0.035 | 3.4 | Oct-1-N | 15.8 | 0.67 | 0.045 | **15** |
| Oct-2-D | 24.3 | 0.2 | 0.026 | 7.8 | Oct-2-N | 18.4 | 0.59 | 0.038 | **16** |
| Oct-3-D | 22.5 | 0.26 | 0.029 | 9.0 | Oct-3-N | 17.7 | 0.59 | 0.040 | **15** |
| Oct-4-D | 21.5 | 0.36 | 0.031 | 12 | Oct-4-N | 17.9 | 0.38 | 0.039 | 10 |
| Oct-5-D | 21.8 | 0.37 | 0.030 | 12 | Oct-5-N | 18.3 | 0.53 | 0.038 | **14** |
| Oct-6-D | 23 | 0.39 | 0.028 | 14 | Oct-6-N | 19.8 | 0.44 | 0.035 | 13 |
| Oct-7-D | 22.6 | 0.47 | 0.029 | 16 | Oct-7-N | 19.1 | 0.46 | 0.036 | 13 |
| Oct-8-D | 17.8 | 0.3 | 0.040 | 7.6 | Oct-8-N | 14.3 | 0.24 | 0.050 | 4.8 |

| | | | | | | | | | |
|---|---|---|---|---|---|---|---|---|---|
| Oct-9-D | 18.7 | 0.43 | 0.037 | 12 | Oct-9-N | 12.5 | 0.08 | 0.057 | 1.4 |
| Oct-10-D | 14.4 | 0.19 | 0.050 | 3.8 | Oct-10-N | 15.3 | 0.26 | 0.047 | **5.5** |
| Oct-11-D | 20.1 | 0.1 | 0.034 | 3.0 | Oct-11-N | 16.6 | 0.38 | 0.043 | 8.9 |
| Oct-12-D | 22.5 | 0.27 | 0.029 | 9.4 | Oct-12-N | 17.8 | 1.22 | 0.040 | **31** |
| Oct-13-D | 23.7 | 0.43 | 0.027 | 16 | Oct-13-N | 17.2 | 0.89 | 0.041 | **22** |
| Oct-14-D | 20.1 | 0.37 | 0.034 | 11 | Oct-14-N | 16.3 | 0.49 | 0.044 | 11 |
| Oct-15-D | 22.5 | 0.34 | 0.029 | 12 | Oct-15-N | 19.7 | 0.33 | 0.035 | 9.5 |

Meanwhile, our data showed that the average $\Delta^{199}$Hg value during the daytime (0.26‰ ± 0.40‰, 1SD, $n = 26$) is (statistically) significantly ($p < 0.05$, t-test) higher than during the nighttime (0.04‰ ± 0.22‰, 1SD, $n = 30$). This slight diel variation of odd-MIF of Hg isotopes was explained in terms of photoreduction of PBM during daytime. In addition we argue that the diel variations of odd-MIF of Hg isotopes does not result from GOM gas-aerosol partitioning. In general, divalent Hg gas-aerosol partitioning is considered as chemisorption and desorption (Rutter and Schauer, 2007). Prior studies showed that the adsorption/desorption and precipitation of aqueous $Hg^{2+}$ had insignificant odd-MIF of Hg isotopes (Jiskra et al., 2012; Smith et al., 2015), suggesting that the GOM partitioning process may not result in the characteristics of odd-MIF of Hg isotopes we observed for the $PM_{2.5}$ samples.

In conclusion, gas-particle partitioning may increase PBM during the night-time due to relatively lower temperature compared to the daytime. The actual increase of PBM during the nighttime may be off-set by lower GOM levels during nighttime when little or no production of GOM by the photo-oxidation of GEM may occur. It is shown that GOM is high during daytime likely due to stronger photo-oxidation. GOM can be adsorbed on to PM where the active Hg species are also photo-reduced to elemental mercury. Such a dynamic and complex adsorption-photoreduction cycle yields lowered PBM levels, along with characteristic Hg isotope properties. In other words, our thesis is that the photochemical reactions cause the concentration reduction of $PM_{2.5}$-Hg as well as fractionation of the $PM_{2.5}$-bound Hg isotope compositions.

To address these issues, we have included a discussion about the possible effects of gas-aerosol partitioning on the diel variation of PBM. See the revised manuscript on line 358 to

370, it reads: "A possible explanation of the observed effects of diel variation of $PM_{2.5}$-Hg would be the temperature-dependent gas-aerosol partitioning of GOM (Rutter and Schauer, 2007; Amos et al., 2012), which favors more adsorption of GOM on PM during nighttime when atmospheric temperature us relatively lower than daytime. However, the magnitude of such adsorption is also proportional to the GOM concentration in the atmosphere. An inverse calculation exercise (in SI) shows that the higher $PM_{2.5}$-Hg measured for our samples would require higher GOM concentrations during the nighttime, which contradicts with prior findings that GOM concentrations are significantly lower during the nighttime than the daytime as GOM is a product of photo-oxidation processes (Poissant et al., 2005; Liu et al., 2007; Amos et al., 2012). In addition, GOM gas-aerosol partitioning is considered a chemisorption and desorption process (Rutter and Schauer, 2007), which unlikely result in appreciable odd-MIF of Hg isotopes (Jiskra et al., 2012; Smith et al., 2015). Therefore, GOM partitioning would have little or no effect on the observed diel variations of $\Delta^{199}$Hg values for $PM_{2.5}$-Hg."

I suggest the authors consider the influence of boundary layer dynamics and stratification: daytime turbulence could lead to mixing of above lying, cleaner free tropospheric air with high MIF, whereas nighttime stratification traps Hg emissions with low MIF."

--Thank you for your suggestion. We agree that the boundary layer was higher during the daytime than at night, and daytime turbulence could help to mix the air between bottom and top of the layer. As your suggest, with constant Hg emission and $PM_{2.5}$ deposition rates, $Hg^{2+}$ photoreduction on $PM_{2.5}$ during the daytime may be enhanced at the top of the boundary layer (up to 1500 m) on a sunny day and produce much more positive odd-MIF of Hg isotopes on $PM_{2.5}$, while at night, the lower boundary layer traps a portion of daytime PBM at much low altitudes (mean of about 300 m). The mixing of residual daytime PBM with newly emitted

PBM in the thinner boundary layer at night may help to explain why nighttime PBM had odd-MIF values closer to source emissions.

Per your suggestions and comments, we have added a discussion about the possible effects of the difference of boundary layer thickness during daytime and nighttime on the diel variations of Hg isotope ratios in the PM$_{2.5}$ samples we collected. See the revised manuscript on line 371 to 381, it reads: "Variation in atmosphere boundary layer height (ABLH) from 1000 to 1300 m during daytime to less than 200 to 300 m during nighttime may have contributed to the diel variation in Hg isotopic composition of PM$_{2.5}$-Hg (Quan et al., 2013). With a high ABLH during daytime, relatively strong turbulence may help mixing the PM$_{2.5}$-Hg from the surface to the upper free troposphere, where photoreactions may be favored due to higher intensities of ultraviolet radiation on clear days. In contrast, a lower ABLH at night may weaken the vertical transport of PM$_{2.5}$-Hg, but enhance the contribution from newly produced PM$_{2.5}$-Hg, possibly resulting in higher concentrations of PM$_{2.5}$-Hg with negative or close to zero $\Delta^{199}$Hg values from emission sources and/or GOM. However, vertically-resolved, day-night measurements of Hg stable isotope ratios in PBM and GOM are needed to fully evaluate the effects of various physical processes on diel variation of the Hg isotopic compositions for the PM$_{2.5}$".

In the current MS submitted to ACP I did not find discussion of gaseous Hg(0) – aerosol partitioning, nor discussion of boundary layer dynamics. The authors should discuss with atmospheric physicists, and see if proxies of boundary layer mixing can be used. For ex. PM2.5 itself seems higher during nighttime than daytime, which is likely due to nighttime boundary layer stratification which traps pollutant emissions. Daytime heating of land and ensuing turbulence will mix boundary layer air with overlying free tropospheric air. Such mixing may, or may not, generate all the trends observed. It should be discussed and counter-argumented.

--We agree that, although PM$_{2.5}$ concentrations had insignificant diel variation ($p$ = 0.887, paired samples $t$-test), the change of boundary layer thickness between daytime and nighttime could affect PBM transformations, and we now address the possibility of this effect as described in our comments above.

In summary, I am convinced that the dataset is novel and of strong interest to the atmospheric Hg community and ACP readers, but I suggest the authors to better think through alternative interpretations, and to respect the reviewing process. The editor and reviewers spend time to try and make your study better.

--Thank you for your suggestions and comments. Although the editor of ES&T did not give us a chance to response your comments, we truly appreciate the editor and reviewers for their comments on this manuscript. We are very glad to have this chance to respond to your comments at here, and we revised the manuscript accordingly.

Minor comments:

L99. "It is intuitive that, while both D and N PM2.5 samples may have similar local or regional sources if the wind trajectory remains unchanged, D samples could have been exposed to more solar radiation than N samples, likely resulting in diel variations in the Hg isotope compositions that are indicative of differences in photochemical transformation of PM2.5-Hg."  Not sure this makes sense: if PM-Hg was emitted 1 week ago and travelled across China, the particles went through 7 day/night periods, all receiving more or less the same amount of radiation.

--We have deleted this sentence.

L211. The statistics of diel variations are discussed here, with reference to Table S3. It appears to me that paired T-tests should be reported in the text, and not means and p values for the whole dataset. A key question is whether PM2.5 shows diel variation in the paired T-test? The p values in the text do not correspond to metrics in Table S3, so the discussion is hard to follow.

--We have revised this paragraph per your suggestions, which now reads (on line 207 to 215 in the revised manuscript): "T-test results (Table S3) showed that diel variation was statistically significant ($p < 0.05$) for Hg contents, $\Delta^{199}$Hg, and $\Delta^{200}$Hg values, as their $p$ values are 0.005, 0.000 and 0.004 resulting from paired samples $t$-test, and are 0.003, 0.017 and 0.019 resulting from independent samples $t$-test. For all samples, Hg contents for D-samples ($0.32 \pm 0.14$ µg g$^{-1}$) were lower than N-samples ($0.48 \pm 0.24$ µg g$^{-1}$), and $\Delta^{199}$Hg and $\Delta^{200}$Hg values for D-samples (mean of 0.26‰ ±0.40‰ and 0.09‰ ± 0.06‰, respectively) were higher than N-samples (−0.04‰ ± 0.22‰ and 0.06‰ ± 0.05‰, respectively). However, PM$_{2.5}$ concentrations and $\delta^{202}$Hg had statistically insignificant ($p > 0.05$) diel variation, as their $p$ values are 0.887 and 0.052 resulting from paired samples $t$-test, and are 0.909 and 0.053 resulting from independent samples $t$-test".

L171. Why 24h back trajectories and not more? What is known about PBM lifetime in the Chinese boundary layer? I think there is a discussion in Horowitz et al., ACP, 2017 on this.

--Per your suggestion, we changed to 72-h back trajectories in the revised manuscript. The results of such 72-h back trajectory frequencies are shown below in Figure R1, indicating that the dominant air masses (over 90%) of source directions estimated from 72-h approach were very similar to those estimated using the 24-h and 48-h approaches.

[Figure]

[Figure]

[Figure]

Figure R1. The frequencies of backward trajectories were calculated for all the samples using the 24h, 48h and 72h Backward Trajectory Model.

L342. "Interestingly, negative D199Hg values in daytime PM2.5-Hg were only observed during a rainy day and an extreme smog event. Scavenging of locally produced GOM during rain or smog events may therefore have contributed to the reversal of the odd-MIF signature of Hg collected as PM2.5 at these times." Why would local GOM have a negative D199? Not clear to reader.

--We have revised these sentences for clarity (see the revised manuscript on line 341 to 346): "Interestingly, negative $\Delta^{199}$Hg values in daytime PM$_{2.5}$-Hg were only observed during a rainy day and an extreme smog event. Since the Hg emitted from local sources had close to zero and negative values of odd-MIF, higher humidity (such as during rainy days) and heavy pollution (the extreme smog) may enhance the effect of scavenging of locally produced gaseous or

particulate Hg during rain or smog events, which may therefore have contributed to the reversal of the odd-MIF signature of Hg collected as $PM_{2.5}$ at these times".

**2 Anonymous Referee #2**

This manuscript quantified the diel variation of Hg isotope composition of particulate bound mercury (PBM) and revealed that daily photochemical reduction of divalent Hg is of critical importance to the fate of PM2.5-Hg in urban atmospheres. The topic is quite interesting and is important for understanding global mercury cycling. Publication is suggested after minor revision.

--Thank you for your comments.

Line 114 Is one air sampler enough? PBM concentration in the air is quite small. To obtain enough mercury for isotope analysis, especially when the sampling time was reduced, it seems that we need more samplers.

--We used one sampler for collecting the PM samples. Among the 61 PM2.5 samples we collected, 56 had sufficient Hg mass for Hg isotope analysis. It would be better if two or more samplers were used simultaneously for sampling so that (1) sufficient mass of PBM can be obtained for isotope analysis and (2) replicates could be used for isotope analysis.

Line 163 Why do you choose the height of 500 m?

--In our back trajectory modeling, we used 500 m as the average boundary layer height in Beijing according to a prior study by (Xiang et al., 2019). The estimated backward HYSPLIT trajectories of air masses should be acceptable. Alternatively, we also used different arrival heights (200, 500, 1000 m above ground level) for estimating the backward trajectories. The

results (see below Figure R2) indicate that the transport pathways are not very sensitive to the selected heights within the studied area.

Per your comments, we have added detail information in the revised version of the Supporting Information.

[Figure]

Figure R2. The frequencies of backward trajectories were calculated for all the samples using the 24h Backward Trajectory Model at three deferent heights of 200, 500 and 1000 m above ground level.

Figure 2 This figure is too busy. Instead listing all data according to time series, is it possible to classify the figure into several subgroup according to the topic you wants to discussed? This figure can be moved to supporting information.

--Per your suggestion, we have revised Figure 2 as following (see below Figure R3), which shows the chronological sequence of MIF ($\Delta^{199}$Hg and $\Delta^{200}$Hg) and MDF ($\delta^{202}$Hg) of the 56 samples collected during the daytime (D, red) and nighttime (N, blue), along with selected weather data including cumulative hours of sunshine (Solar) and air mass backward-trajectory directions.

[Figure]

Figure R3. Chronological sequence of MIF ($\Delta^{199}$Hg and $\Delta^{200}$Hg) and MDF ($\delta^{202}$Hg) of the 56 samples collected during the daytime (D, red) and nighttime (N, blue), along with selected

weather data including cumulative hours of sunshine (Solar) and air mass backward-trajectory directions.

Figure 2(a) How to explain the negative value of $\Delta^{199}$Hg on Sep 28?

--The explanation had been described in Line 363 to 373. "Interestingly, negative $\Delta^{199}$Hg values in daytime PM$_{2.5}$-Hg were only observed during a rainy day and an extreme smog event. Since the Hg emitted from local sources had close to zero and negative values of odd-MIF, higher humidity (such as during rainy days) and heavy pollution (the extreme smog) may enhance the effect of scavenging of locally produced gaseous or particulate Hg during rain or smog events, which may therefore have contributed to the reversal of the odd-MIF signature of Hg collected as PM$_{2.5}$ at these times. In addition, the negative $\Delta^{199}$Hg values in PM$_{2.5}$ may have resulted from the contribution of biomass burning with limited photoreduction effect during periods of less sunshine (Fig. 2 and Table S1) since plant foliage has negative $\Delta^{199}$Hg values (Yu et al., 2016) and more negative $\Delta^{199}$Hg values (down to −0.53‰) of PM$_{2.5}$-Hg in Beijing were related to biomass burning, a source of PM$_{2.5}$-Hg south of Beijing in autumn (Huang et al., 2016)."

Figure 2(f) all legends are suggested to be listed on the top of this figure. It is difficult to find "clear" "cloudy" "rain" in this figure.

--We have revised the legends in the revised Figure 2(d).

Line 356 "While our results cannot exclude the effects of other possible processes, such as oxidation, adsorption (and desorption), and precipitation, based on the limited previous studies (Jiskra et al., 2012; Smith et al., 2015; Sun et al., 2016), these processes are not likely to be

important to the diel variation of odd-MIF of Hg isotopes in PM2.5-Hg we observed." The observed isotope fractionation is a phenomenon while the photochemical reduction is one process leading to this phenomenon. How can you exclude the impact from other processes? Evidences are required to prove this conclusion.

--We agree that these observations may be the result of multiple processes. We now address the possible contributions of two additional physical process, adsorption of GOM to PM and reduced boundary layer mixing of PBM at night (see responses to Referee 1).

Figure 5(a) What is the main reason that caused the variation of Δ199Hg during the night time? Is it possible caused by measurement error? If this is true, it is better to point out this in method part.

--Our data showed that the $\Delta^{199}$Hg values ranged from 0.01‰ to 0.30‰ for the nighttime samples in Figure 5(a) and ranged from -0.51‰ to 0.55‰ for all nighttime samples in this study. The method we used for quantifying Hg isotopes bears an uncertainty (2SD) of 0.06‰ for $\Delta^{199}$Hg for the samples. Statistically, differences between $\Delta^{199}$Hg values for daytime and nighttime samples were clearly significant and should not have been caused by uncertainty of the method. To help readers understand this issue, we have added the measurement uncertainty in the caption of Figure 5.

extract Hg bound to PM samples, each filter was rolled into a cylinder and placed in a sample

quartz tube. Both ends of the tube were capped with quartz wool (pre-cleaned at 500°C) to

prevent particle emission. Each tube was combusted over 2 h in a temperature-programmed

dual-stage quartz tube combustion furnace in which the temperature of the first furnace was

60   incrementally increased to 900°C whereas the second furnace was held at 950°C. The resulting

Hg vapor was swept by $O_2$ gas (Hg free) into the 40% acid trapping solution. The trapping

solution was diluted with Milli-Q $H_2O$ to 10 mL to a final acid concentration of 20%. The

accuracy and precision of the dual-stage combustion protocol were evaluated by the analysis of

the GBW07405 using the same digestion method. The detection limit given by the procedural

65   blanks (< 0.3 ng) for this dual-stage combustion method was negligibly low compared to the

total Hg mass ($\geq$ 10 ng) extracted from either $PM_{2.5}$ samples or procedural standards.

[Figure]

Schematic diagram of the combustion-trapping assembly from Huang et al. (2015).

70   **1.3   Mercury concentration and stable isotope composition measurements**

The methods used to measure the Hg content and isotope ratio were published elsewhere

(Huang et al., 2015). In brief, a small fraction of each trapping solution (20% acid mixture) was

used to measure the Hg content on cold-vapor atomic fluorescence spectroscopy (CVAFS,

Tekran 2500, Tekran® Instruments Corporation, CA), with a precision better than 10%. The

75  recoveries of Hg for the standard GBW07405 were in the acceptable range of 95 to 105% with

an average value of 98% (1 SD = 6%, $n$ = 6); but no recovery of Hg for the $PM_{2.5}$ samples was

determined due to limited availability of the samples.

A total of 61 $PM_{2.5}$ samples were collected during the sampling campaign.    After analysis

of Hg contents, we found that 56 $PM_{2.5}$ samples (including 26 daytime and 30 nighttime samples)

80  have sufficient Hg mass and hence were further analyzed for Hg isotope compositions using a

multicollector inductively coupled plasma mass spectrometer (MC-ICP-MS, Nu Instruments

Ltd., UK) equipped with a continuous flow cold vapor generation system. Detailed protocols

for the Hg isotope analysis can be found in Huang et al. (2015). The Faraday cups were

positioned to simultaneously collect five Hg isotopes and two Tl isotopes including $^{205}$Tl (H3),

85  $^{203}$Tl (H1), $^{202}$Hg (Ax), $^{201}$Hg (L1), $^{200}$Hg (L2), $^{199}$Hg (L3), and $^{198}$Hg (L4). $^{196}$Hg and $^{204}$Hg

were not measured due to their very low abundance. Instrumental mass bias was corrected using

an internal standard (NIST SRM 997 Tl) and strict sample-standard bracketing with NIST SRM

3133 Hg standard. For quality assurance and control, the well-known reference material UM-

Almaden and the NIST SRM 3177 Hg were inserted repeatedly into the sampling list after every

90  ten and five real samples, measured regularly during sample analysis session, and calibrated

periodically against the NIST SRM 3133 Hg as well as samples.

Delta ($\delta$) notation is used to represent MDF in units of per mil (‰) as defined by the

following equation (Blum and Bergquist, 2007):

$$\delta^{x}Hg \text{ (‰)} = [(^{x}Hg/^{198}Hg)_{sample}/(^{x}Hg/^{198}Hg)_{NIST3133} - 1] \times 1000 \qquad (1)$$

95  where x = 199, 200, 201, and 202. MIF is reported as the deviation of a measured delta value

from the theoretically predicted MDF value according to the equation:

$$\Delta^x Hg\ (‰) = \delta^x Hg - \beta \times \delta^{202}Hg \qquad\qquad (2)$$

where the mass-dependent scaling factor $\beta$ is 0.252, 0.5024, and 0.752 for $^{199}$Hg, $^{200}$Hg and $^{201}$Hg, respectively (Blum and Bergquist, 2007).

100

**1.4 Backward trajectory analysis**

The backward HYSPLIT trajectories of air masses at a height of 500 m above ground level and arriving at the sampling site (at 39.9725 N 116.3683 E) were simulated. Because the average boundary layer heights was reported about 500 m in Beijing (Xiang et al., 2019), so the

105 arrival height of 500 m used for backward HYSPLIT trajectories of air masses could be acceptable in this study. In fact, different arrival heights (200, 500, 1000 m AGL) of backward trajectories were tested and results indicated that the transport pathways were not very sensitive to the selected heights within the studied area. Backward trajectories for each sample, ending at 1100 UTC (equal to local time 7:00 p.m.) for daytime sample and ending at 2300 UTC (equal

110 to local time 7:00 a.m.) for nighttime sample, were calculated every 1 hrs using the Internet-Based HYSPLIT Trajectory Model and gridded meteorological data (Global Data Assimilation System, GDAS1) from the U.S. National Oceanic and Atmospheric Administration (NOAA) and were shown below (Fig. S1). The obtained average directions of arriving air masses for each sample were summarized in Table S1. The frequencies of backward trajectories were also

115 calculated for all the samples taken during Sept. 15[th] to Oct. 16[th] 2015 using the Internet-Based HYSPLIT Trajectory Model and the archived GDAS0p5, with an interval of 3 hrs, each trajectory total run time 72 hrs and a 0.5 × 0.5 degree trajectory frequency grid resolution. The results of such simulation showed the dominating air mass arriving from southwest of the sampling site (see Fig. 1).

   **1.5   GOM calculation**

We used an inverse approach and a GOM partitioning model to compute hypothetic GOM

levels corresponding to each of our PBM observations at ambient temperature. We used the

GOM gas-aerosol partitioning model proposed by Amos et al. (2012), which has the following

equation: $\log_{10}(K^{-1}) = (10\pm1) - (2500\pm300)/T$, where $K = (\text{PBM}/\text{PM}_{2.5})/\text{GOM}$ with PBM and

125   GOM in common volumetric units (pg m$^{-3}$), PM$_{2.5}$ in µg m$^{-3}$, and $T$ in K. We used the measured

PM$_{2.5}$-Hg as PBM and assumed that the PM$_{2.5}$-Hg measured for each sample is 100% in divalent

and active mercury forms. The calculated GOM concentrations are presented in the following

Table S4. In summary, the calculated GOM concentrations range from 1.5 to 31 pg m$^{-3}$, with

average values of 11±5 pg m$^{-3}$ during the daytime and 13±7 pg m$^{-3}$ during the nighttime.

130   Overall, the calculated GOM exhibit insignificant ($p$ = 0195, paired samples $t$-test) diel

variation of GOM concentration, i.e., there would be little or no difference of GOM between

day- and night-time. Close inspection of the data (Table S4) showed that half of the paired daynight samples have higher calculated GOM concentrations during the nighttime than in daytime.

135 **Table S1.** List of 61 PM$_{2.5}$ samples and their associated weather data.

| Name | Sampling date | Start time | End time | Directions of arriving air mass | Weather | Sunshine duration (hrs) | SH (MJ/m$^2$) | O$_3$ (ppbv) | T (°C) | RH (%) | MWS (m/s) | WS (m/s) |
|---|---|---|---|---|---|---|---|---|---|---|---|---|
| Sept-15-N | Sept-15-2015 | 19:02 | 7:02 | S-SW | Cloudy | | | 7.3 | 19.8 | 68 | 3.9 | 2 |
| Sept-16-D | Sept-16-2015 | 8:25 | 18:55 | SW | Sunny | 8 | 6.40 | 67.7 | 24.1 | 52 | 3.9 | 2 |
| Sept-16-N | Sept-16-2015 | 19:02 | 7:02 | SW | Cloudy | | | 5.6 | 21.1 | 70 | 3.9 | 2 |
| Sept-17-D | Sept-17-2015 | 8:13 | 18:43 | SW | Cloudy | 4 | 3.02 | 72.3 | 24.8 | 56 | 4.0 | 2 |
| Sept-17-N | Sept-17-2015 | 19:03 | 7:33 | SW-NW | Cloudy+Rain | | | 19.1 | 22.6 | 74 | 4.0 | 2 |
| Sept-18-D | Sept-18-2015 | 8:19 | 18:49 | N | Sunny | 9 | 13.24 | 43.2 | 27.3 | 39 | 3.1 | 2 |
| Sept-18-N | Sept-18-2015 | 18:57 | 7:27 | N-NE | Clear | | | 1.1 | 21.7 | 54 | 3.1 | 2 |
| Sept-19-D | Sept-19-2015 | 8:07 | 18:37 | NE-E | Sunny | 11 | 17.20 | 41.5 | 26.1 | 33 | 3.6 | 1 |
| Sept-19-N | Sept-19-2015 | 18:48 | 7:18 | S | Clear | | | 4.8 | 21.7 | 60 | 3.6 | 1 |
| Sept-20-D | Sept-20-2015 | 8:04 | 18:34 | SW | Cloudy | 7 | 4.45 | 34.4 | 24.5 | 55 | 4.3 | 2 |
| Sept-20-N | Sept-20-2015 | 18:42 | 7:12 | SW | Cloudy | | | 5.7 | 21.9 | 62 | 4.3 | 2 |
| Sept-21-D | Sept-21-2015 | 8:17 | 18:47 | SW | Sunny | 9 | 6.04 | 41.3 | 25.2 | 49 | 4.1 | 2 |
| Sept-21-N | Sept-21-2015 | 18:55 | 7:25 | S | Cloudy | | | 18.6 | 22.6 | 62 | 4.1 | 2 |
| Sept-22-D | Sept-22-2015 | 8:18 | 18:18 | S-SW | Overcast+Rain | 0 | 0.43 | 31.4 | 22.9 | 70 | 5.4 | 2 |
| Sept-22-N | Sept-22-2015 | 18:28 | 6:58 | W-NW | Cloudy | | | 6.3 | 18.3 | 86 | 5.4 | 2 |
| Sept-23-D | Sept-23-2015 | 8:13 | 18:13 | S-NW | Sunny | 9 | 11.20 | 36.1 | 23.9 | 54 | 2.7 | 2 |
| Sept-23-N | Sept-23-2015 | 18:13 | 6:56 | S-SW | Cloudy | | | 3.3 | 21.6 | 71 | 2.7 | 2 |
| Sept-24-D | Sept-24-2015 | 8:07 | 18:07 | S | Cloudy | 2 | 0.73 | 26.5 | 23.0 | 72 | 3.8 | 2 |
| Sept-24-N | Sept-24-2015 | 18:15 | 6:45 | SE-W | Cloudy+Rain | | | 19.4 | 17.7 | 87 | 3.8 | 2 |
| Sept-25-D | Sept-25-2015 | 8:28 | 18:28 | NW | Sunny | 11 | 18.43 | 28.6 | 24.4 | 19 | 4.8 | 2 |
| Sept-25-N | Sept-25-2015 | 19:03 | 7:33 | SW-NW | Clear | | | 1.2 | 17.3 | 44 | 4.8 | 2 |
| Sept-26-D | Sept-26-2015 | 8:06 | 18:06 | SW | Sunny | 10 | 13.90 | 36.1 | 23.7 | 37 | 5.9 | 2 |
| Sept-26-N | Sept-26-2015 | 18:12 | 6:42 | SW-W | Clear | | | 6.2 | 20.5 | 59 | 5.9 | 2 |
| Sept-27-D | Sept-27-2015 | 8:21 | 18:21 | N-NE | Sunny | 10 | 12.56 | 41.9 | 23.8 | 42 | 3.3 | 2 |
| Sept-27-N | Sept-27-2015 | 18:38 | 7:08 | N-E | Cloudy | | | 10.2 | - | - | 3.3 | 2 |
| Sept-28-D | Sept-28-2015 | 8:39 | 18:39 | E | Overcast+Rain | 0 | - | 14.1 | 18.1 | 82 | 4.0 | 2 |
| Sept-28-N | Sept-28-2015 | 18:47 | 7:17 | E | Overcast+Rain | | | 1.0 | 17.4 | 81 | 4.0 | 2 |
| Sept-29-D | Sept-29-2015 | 8:03 | 18:03 | E-SE | Overcast+Rain | 0 | - | 6.7 | 15.7 | 88 | 3.0 | 2 |
| Sept-29-N | Sept-29-2015 | 18:33 | 7:03 | SE | Overcast+Rain | | | 0.9 | 14.7 | 92 | 3.0 | 2 |
| Sept-30-D | Sept-30-2015 | 8:14 | 18:14 | SW | Cloudy+Rain | 3 | 3.70 | 19.4 | 18.1 | 68 | 3.2 | 2 |
| Sept-30-N | Sept-30-2015 | 18:55 | 7:25 | SW-NW | Cloudy+Rain | | | 13.9 | 15.5 | 67 | 3.2 | 2 |
| Oct-1-D | Oct-1-2015 | 7:26 | 18:31 | NW | Sunny | 11 | 17.68 | 29.2 | 19.4 | 27 | 7.4 | 3 |
| Oct-1-N | Oct-1-2015 | 18:41 | 7:11 | NW | Clear | | | 1.7 | 15.8 | 51 | 7.4 | 3 |
| Oct-2-D | Oct-2-2015 | 8:50 | 18:50 | NW | Sunny | 11 | 16.00 | 37.6 | 24.3 | 27 | 4.8 | 2 |
| Oct-2-N | Oct-2-2015 | 18:58 | 7:28 | NW | Clear | | | 3.2 | 18.4 | 49 | 4.8 | 2 |
| Oct-3-D | Oct-3-2015 | 8:05 | 18:05 | N-S | Sunny | 10 | 14.35 | 29.1 | 22.5 | 35 | 4.6 | 2 |
| Oct-3-N | Oct-3-2015 | 18:30 | 7:00 | SW | Clear | | | 2.1 | 17.7 | 66 | 4.6 | 2 |
| Oct-4-D | Oct-4-2015 | 8:20 | 18:20 | SW | Sunny | 9 | 6.40 | 32.1 | 21.5 | 52 | 2.7 | 1 |
| Oct-4-N | Oct-4-2015 | 18:44 | 7:14 | SW | Clear | | | 1.4 | 17.9 | 74 | 2.7 | 1 |
| Oct-5-D | Oct-5-2015 | 8:03 | 18:03 | SW | Haze | 6 | 3.89 | 51.6 | 21.8 | 59 | 3.0 | 1 |
| Oct-5-N | Oct-5-2015 | 19:04 | 7:34 | SW | Haze | | | 2.8 | 18.3 | 83 | 3.0 | 1 |
| Oct-6-D | Oct-6-2015 | 8:24 | 18:24 | SW-W | Haze | 5 | 3.03 | 71.4 | 23.0 | 60 | 2.7 | 1 |
| Oct-6-N | Oct-6-2015 | 19:25 | 7:25 | SW | Haze | | | 3.4 | 19.8 | 81 | 2.7 | 1 |
| Oct-7-D | Oct-7-2015 | 7:55 | 16:55 | SW-NW | Haze | 2 | 1.66 | 31.6 | 22.6 | 68 | 3.4 | 2 |
| Oct-7-N | Oct-7-2015 | 18:00 | 6:00 | NW | Clear | | | 18.9 | 19.1 | 24 | 3.4 | 2 |
| Oct-8-D | Oct-8-2015 | 8:07 | 18:07 | NW | Sunny | 11 | 16.11 | 24.9 | 17.8 | 15 | 7.1 | 3 |
| Oct-8-N | Oct-8-2015 | 18:43 | 6:43 | NW | Clear | | | 2.6 | 14.3 | 31 | 7.1 | 3 |
| Oct-9-D | Oct-9-2015 | 7:43 | 17:13 | NW | Sunny | 11 | 12.81 | 18.1 | 18.7 | 20 | 8.3 | 4 |
| Oct-9-N | Oct-9-2015 | 17:53 | 5:23 | NW-N | Cloudy | | | 17.9 | 12.5 | 30 | 8.3 | 4 |
| Oct-10-D | Oct-10-2015 | 8:10 | 18:10 | NW-N | Sunny | 10 | 12.75 | 23.8 | 14.4 | 24 | 7.7 | 4 |
| Oct-10-N | Oct-10-2015 | 18:38 | 7:08 | N | Cloudy | | | 23.3 | 15.3 | 27 | 7.7 | 4 |
| Oct-11-D | Oct-11-2015 | 8:35 | 18:00 | N | Sunny | 10 | 12.63 | 30.4 | 20.1 | 26 | 7.1 | 3 |
| Oct-11-N | Oct-11-2015 | 18:08 | 6:38 | N | Cloudy | | | 5.8 | 16.6 | 27 | 7.1 | 3 |
| Oct-12-D | Oct-12-2015 | 7:49 | 17:31 | NW | Sunny | 11 | 14.67 | 27.4 | 22.5 | 20 | 5.9 | 1 |
| Oct-12-N | Oct-12-2015 | 17:40 | 6:10 | NW | Cloudy | | | 2.7 | 17.8 | 34 | 5.9 | 1 |
| Oct-13-D | Oct-13-2015 | 8:26 | 17:48 | NW-W | Sunny | 11 | 13.84 | 22.4 | 23.7 | 22 | 3.7 | 1 |
| Oct-13-N | Oct-13-2015 | 17:53 | 6:23 | SW-SE | Cloudy | | | 1.3 | 17.2 | 50 | 3.7 | 1 |
| Oct-14-D | Oct-14-2015 | 8:17 | 17:47 | S-E | Cloudy | 4 | 2.99 | 12.4 | 20.1 | 45 | 3.5 | 2 |
| Oct-14-N | Oct-14-2015 | 17:53 | 6:23 | SW-NW | Cloudy | | | 1.1 | 16.3 | 64 | 3.5 | 2 |
| Oct-15-D | Oct-15-2015 | 8:28 | 17:33 | W-NW | Sunny | 9 | 8.31 | 31.9 | 22.5 | 38 | 3.1 | 2 |
| Oct-15-N | Oct-15-2015 | 17:39 | 6:09 | W | Cloudy | | | 3.3 | 19.7 | 55 | 3.1 | 2 |

SH is the daily solar radiation on a horizontal surface, T is 12-hour averaged temperature, RH is 12-hour averaged relative humidity, MWS is the daily (24-hour) maximum wind speed, and WS is the daily average wind speed.

**Table S2.** Contents of PM$_{2.5}$, Hg in PM$_{2.5}$ (PM$_{2.5}$-Hg) and Hg isotopic composition of PM$_{2.5}$-Hg.

| Name | PM$_{2.5}$ (μg/m³) | Hg Con. (μg/g) | δ$^{202}$Hg (‰) | 2SD | Δ$^{199}$Hg (‰) | 2SD | Δ$^{200}$Hg (‰) | 2SD | Δ$^{201}$Hg (‰) | 2SD |
|---|---|---|---|---|---|---|---|---|---|---|
| Sept-15-N | 85 | 0.52 | -0.89 | 0.14 | 0.05 | 0.06 | 0.13 | 0.04 | -0.08 | 0.07 |
| Sept-16-D | 71 | 0.41 | -0.61 | 0.14 | 0.30 | 0.06 | 0.06 | 0.04 | 0.05 | 0.07 |
| Sept-16-N | 88 | 0.44 | -0.53 | 0.14 | -0.05 | 0.06 | 0.06 | 0.04 | -0.11 | 0.07 |
| Sept-17-D | 76 | 0.38 | -0.47 | 0.14 | 0.21 | 0.06 | 0.01 | 0.04 | 0.08 | 0.07 |
| Sept-17-N | 94 | 0.47 | -0.27 | 0.14 | -0.08 | 0.06 | 0.03 | 0.04 | -0.15 | 0.07 |
| Sept-18-D | 32 | 0.17 | -0.72 | 0.14 | 0.90 | 0.06 | 0.08 | 0.04 | 0.64 | 0.07 |
| Sept-18-N | 43 | 0.31 | -1.29 | 0.14 | 0.04 | 0.06 | -0.02 | 0.04 | -0.12 | 0.09 |
| Sept-19-D | 13 | 0.09 | | | | | | | | |
| Sept-19-N | 52 | 0.29 | -0.98 | 0.14 | 0.09 | 0.06 | 0.02 | 0.04 | -0.06 | 0.07 |
| Sept-20-D | 63 | 0.61 | -0.48 | 0.14 | 0.16 | 0.06 | 0.12 | 0.04 | 0.02 | 0.07 |
| Sept-20-N | 63 | 0.89 | -0.59 | 0.14 | 0.01 | 0.06 | 0.04 | 0.04 | 0.06 | 0.07 |
| Sept-21-D | 61 | 0.48 | -0.34 | 0.14 | 0.50 | 0.06 | 0.05 | 0.04 | 0.34 | 0.07 |
| Sept-21-N | 83 | 0.62 | -0.69 | 0.14 | 0.10 | 0.06 | 0.06 | 0.04 | 0.07 | 0.07 |
| Sept-22-D | 95 | 0.53 | -0.40 | 0.14 | 0.28 | 0.06 | 0.03 | 0.04 | 0.18 | 0.07 |
| Sept-22-N | 23 | 0.31 | -0.83 | 0.14 | -0.06 | 0.06 | -0.01 | 0.04 | -0.09 | 0.07 |
| Sept-23-D | 19 | 0.15 | | | | | | | | |
| Sept-23-N | 39 | 0.54 | -0.87 | 0.14 | 0.17 | 0.06 | 0.03 | 0.04 | 0.09 | 0.07 |
| Sept-24-D | 84 | 0.38 | -0.25 | 0.14 | 0.02 | 0.06 | 0.11 | 0.04 | 0.03 | 0.07 |
| Sept-24-N | 47 | 0.20 | -0.47 | 0.14 | 0.05 | 0.06 | 0.06 | 0.04 | -0.11 | 0.07 |
| Sept-25-D | 9 | 0.38 | -0.49 | 0.14 | 0.21 | 0.06 | 0.18 | 0.04 | 0.21 | 0.07 |
| Sept-25-N | 33 | 0.14 | -0.57 | 0.14 | 0.42 | 0.06 | 0.08 | 0.04 | 0.27 | 0.07 |
| Sept-26-D | 24 | 0.20 | -0.37 | 0.14 | 1.04 | 0.06 | 0.10 | 0.04 | 0.71 | 0.07 |
| Sept-26-N | 51 | 0.44 | -0.64 | 0.18 | 0.30 | 0.06 | 0.09 | 0.04 | 0.17 | 0.07 |
| Sept-27-D | 31 | 0.39 | -0.30 | 0.14 | 0.76 | 0.06 | 0.12 | 0.04 | 0.61 | 0.07 |
| Sept-27-N | 46 | 0.78 | -0.62 | 0.14 | 0.15 | 0.06 | 0.07 | 0.04 | 0.08 | 0.07 |
| Sept-28-D | 34 | 0.32 | -0.38 | 0.14 | -0.48 | 0.06 | 0.02 | 0.04 | -0.52 | 0.07 |
| Sept-28-N | 34 | 0.34 | -0.32 | 0.14 | -0.46 | 0.06 | 0.01 | 0.04 | -0.45 | 0.07 |
| Sept-29-D | 52 | 0.48 | 0.29 | 0.14 | 0.06 | 0.06 | 0.20 | 0.04 | 0.26 | 0.09 |
| Sept-29-N | 13 | 0.36 | -0.82 | 0.14 | -0.04 | 0.06 | 0.02 | 0.04 | -0.03 | 0.07 |
| Sept-30-D | 14 | 0.16 | -0.23 | 0.14 | -0.13 | 0.06 | 0.16 | 0.04 | -0.14 | 0.07 |
| Sept-30-N | 22 | 0.64 | -0.26 | 0.14 | -0.04 | 0.06 | 0.08 | 0.04 | -0.05 | 0.07 |
| Oct-1-D | 7 | 0.12 | | | | | | | | |
| Oct-1-N | 19 | 0.67 | -0.91 | 0.14 | 0.11 | 0.06 | 0.11 | 0.04 | 0.09 | 0.07 |
| Oct-2-D | 18 | 0.20 | -0.54 | 0.14 | 0.29 | 0.06 | 0.14 | 0.04 | 0.26 | 0.07 |
| Oct-2-N | 31 | 0.59 | -1.49 | 0.14 | 0.13 | 0.06 | -0.02 | 0.04 | 0.18 | 0.07 |
| Oct-3-D | 19 | 0.26 | -0.21 | 0.14 | 0.86 | 0.06 | 0.21 | 0.04 | 0.59 | 0.07 |
| Oct-3-N | 39 | 0.59 | -0.95 | 0.14 | 0.18 | 0.06 | 0.07 | 0.04 | 0.20 | 0.07 |
| Oct-4-D | 88 | 0.36 | -0.80 | 0.14 | 0.27 | 0.06 | 0.02 | 0.04 | 0.09 | 0.07 |
| Oct-4-N | 119 | 0.38 | -0.97 | 0.14 | -0.11 | 0.06 | 0.02 | 0.04 | -0.16 | 0.07 |
| Oct-5-D | 114 | 0.37 | -0.32 | 0.14 | -0.53 | 0.06 | 0.09 | 0.04 | -0.64 | 0.07 |
| Oct-5-N | 138 | 0.53 | -0.69 | 0.14 | -0.51 | 0.06 | 0.06 | 0.04 | -0.54 | 0.07 |
| Oct-6-D | 156 | 0.39 | -0.09 | 0.14 | -0.40 | 0.06 | 0.08 | 0.04 | -0.57 | 0.07 |
| Oct-6-N | 158 | 0.44 | 0.16 | 0.14 | -0.15 | 0.06 | 0.10 | 0.04 | -0.12 | 0.07 |
| Oct-7-D | 138 | 0.47 | 0.20 | 0.14 | 0.69 | 0.06 | 0.08 | 0.04 | 0.42 | 0.07 |
| Oct-7-N | 128 | 0.46 | 0.52 | 0.14 | 0.55 | 0.06 | 0.14 | 0.04 | 0.39 | 0.07 |
| Oct-8-D | 4 | 0.30 | -0.22 | 0.14 | 0.32 | 0.06 | 0.19 | 0.04 | 0.20 | 0.07 |
| Oct-8-N | 16 | 0.24 | 0.55 | 0.14 | -0.07 | 0.06 | 0.09 | 0.04 | 0.01 | 0.07 |
| Oct-9-D | 24 | 0.43 | -0.47 | 0.14 | 0.04 | 0.06 | 0.07 | 0.04 | 0.12 | 0.07 |
| Oct-9-N | 17 | 0.08 | | | | | | | | |
| Oct-10-D | 14 | 0.19 | -0.82 | 0.14 | 0.33 | 0.06 | 0.08 | 0.04 | 0.55 | 0.07 |
| Oct-10-N | 8 | 0.26 | -0.61 | 0.14 | 0.32 | 0.06 | 0.11 | 0.04 | 0.28 | 0.07 |
| Oct-11-D | 12 | 0.10 | | | | | | | | |
| Oct-11-N | 15 | 0.38 | -0.42 | 0.14 | 0.17 | 0.06 | 0.14 | 0.04 | 0.20 | 0.07 |
| Oct-12-D | 10 | 0.27 | -0.79 | 0.14 | 0.28 | 0.06 | 0.12 | 0.04 | 0.27 | 0.07 |
| Oct-12-N | 26 | 1.22 | -0.90 | 0.14 | -0.03 | 0.06 | 0.05 | 0.04 | -0.03 | 0.07 |
| Oct-13-D | 19 | 0.43 | -0.70 | 0.14 | 0.23 | 0.06 | 0.01 | 0.04 | 0.16 | 0.07 |
| Oct-13-N | 60 | 0.89 | -0.80 | 0.14 | -0.01 | 0.06 | 0.06 | 0.04 | -0.06 | 0.07 |
| Oct-14-D | 50 | 0.37 | -0.78 | 0.14 | 0.01 | 0.06 | -0.01 | 0.04 | 0.00 | 0.07 |
| Oct-14-N | 82 | 0.49 | -1.21 | 0.14 | -0.20 | 0.06 | -0.02 | 0.04 | -0.18 | 0.07 |
| Oct-15-D | 50 | 0.34 | -0.60 | 0.14 | 0.57 | 0.06 | 0.08 | 0.04 | 0.52 | 0.07 |
| Oct-15-N | 95 | 0.33 | -0.37 | 0.14 | 0.21 | 0.06 | 0.07 | 0.04 | 0.33 | 0.07 |

**Table S3.** The below results of Paired Samples T-Test and Independent Samples T-Test were obtained using the IBM SPSS Statistics Version 22. The paired samples were consecutive day and night PM$_{2.5}$ samples, for example, Sept-16-D and Sept-16-N were paired samples.

Paired Samples Test

| Day - Night | Paired Differences | | | | | t | df | Sig. (2-tailed) |
| --- | --- | --- | --- | --- | --- | --- | --- | --- |
| | Mean | Std. Deviation | Std. Error Mean | 95% Confidence Interval of the Difference | | | | |
| | | | | Lower | Upper | | | |
| PM$_{2.5}$ | -2.06667 | 78.67959 | 14.36486 | -31.44611 | 27.31278 | -0.144 | 29 | 0.887 |
| Hg Con. | -0.15263 | 0.27183 | 0.04963 | -0.25414 | -0.05113 | -3.075 | 29 | 0.005 |
| $\delta^{202}$Hg | 0.16960 | 0.41413 | 0.08283 | -0.00134 | 0.34054 | 2.048 | 24 | 0.052 |
| $\Delta^{199}$Hg | 0.24400 | 0.28384 | 0.05677 | 0.12684 | 0.36116 | 4.298 | 24 | 0.000 |
| $\Delta^{200}$Hg | 0.04120 | 0.06412 | 0.01282 | 0.01473 | 0.06767 | 3.213 | 24 | 0.004 |

145

Independent Samples Test

| Day vs Night | Mean Difference | Std. Error Difference | 95% Confidence Interval of the Difference | | t | df | Sig. (2-tailed) |
| --- | --- | --- | --- | --- | --- | --- | --- |
| | | | Lower | Upper | | | |
| PM$_{2.5}$ | -2.57742 | 22.46315 | -47.52607 | 42.37123 | -0.115 | 59 | 0.909 |
| Hg Con. | -0.15408 | 0.04966 | -0.25393 | -0.05423 | -3.103 | 47.858 | 0.003 |
| $\delta^{202}$Hg | 0.20549 | 0.10385 | -0.00272 | 0.41370 | 1.979 | 54 | 0.053 |
| $\Delta^{199}$Hg | 0.21982 | 0.08777 | 0.04209 | 0.39755 | 2.505 | 37.666 | 0.017 |
| $\Delta^{200}$Hg | 0.03464 | 0.01437 | 0.00582 | 0.06346 | 2.410 | 54 | 0.019 |

**Table S4**. Calculated GOM concentrations of day and night samples. The value of GOM concentrations higher at night than the consecutively days are in bold text.

| Daytime samples | Ave. T (°C) | Hg Con. ($\mu$g g$^{-1}$) | $K$ m$^3$ $\mu$g$^{-1}$ | GOM pg m$^{-3}$ | Nighttime samples | Ave. T (°C) | Hg Con. ($\mu$g g$^{-1}$) | $K$ m$^3$ $\mu$g$^{-1}$ | GOM pg m$^{-3}$ |
|---|---|---|---|---|---|---|---|---|---|
| | | | | | Sept-15-N | 19.8 | 0.52 | 0.035 | 15 |
| Sept-16-D | 24.1 | 0.41 | 0.026 | 16 | Sept-16-N | 21.1 | 0.44 | 0.032 | 14 |
| Sept-17-D | 24.8 | 0.38 | 0.025 | 15 | Sept-17-N | 22.6 | 0.47 | 0.029 | **16** |
| Sept-18-D | 27.3 | 0.17 | 0.021 | 8.0 | Sept-18-N | 21.7 | 0.31 | 0.030 | **10** |
| Sept-19-D | 26.1 | 0.09 | 0.023 | 3.9 | Sept-19-N | 21.7 | 0.29 | 0.030 | 10 |
| Sept-20-D | 24.5 | 0.61 | 0.025 | 24 | Sept-20-N | 21.9 | 0.89 | 0.030 | **30** |
| Sept-21-D | 25.2 | 0.48 | 0.024 | 20 | Sept-21-N | 22.6 | 0.62 | 0.029 | **22** |
| Sept-22-D | 22.9 | 0.53 | 0.028 | 19 | Sept-22-N | 18.3 | 0.31 | 0.038 | 8.1 |
| Sept-23-D | 23.9 | 0.15 | 0.026 | 5.7 | Sept-23-N | 21.6 | 0.54 | 0.031 | **18** |
| Sept-24-D | 23 | 0.38 | 0.028 | 14 | Sept-24-N | 17.7 | 0.2 | 0.040 | 5.0 |
| Sept-25-D | 24.4 | 0.38 | 0.025 | 15 | Sept-25-N | 17.3 | 0.14 | 0.041 | 3.4 |
| Sept-26-D | 23.7 | 0.2 | 0.027 | 7.5 | Sept-26-N | 20.5 | 0.44 | 0.033 | 13 |
| Sept-27-D | 23.8 | 0.39 | 0.026 | 15 | Sept-27-N | - | 0.78 | | |
| Sept-28-D | 18.1 | 0.32 | 0.039 | 8 | Sept-28-N | 17.4 | 0.34 | 0.041 | 8.4 |
| Sept-29-D | 15.7 | 0.48 | 0.046 | 11 | Sept-29-N | 14.7 | 0.36 | 0.049 | 7.4 |
| Sept-30-D | 18.1 | 0.16 | 0.039 | 4.1 | Sept-30-N | 15.5 | 0.64 | 0.046 | **14** |
| Oct-1-D | 19.4 | 0.12 | 0.035 | 3.4 | Oct-1-N | 15.8 | 0.67 | 0.045 | **15** |
| Oct-2-D | 24.3 | 0.2 | 0.026 | 7.8 | Oct-2-N | 18.4 | 0.59 | 0.038 | **16** |
| Oct-3-D | 22.5 | 0.26 | 0.029 | 9.0 | Oct-3-N | 17.7 | 0.59 | 0.040 | **15** |
| Oct-4-D | 21.5 | 0.36 | 0.031 | 12 | Oct-4-N | 17.9 | 0.38 | 0.039 | 10 |
| Oct-5-D | 21.8 | 0.37 | 0.030 | 12 | Oct-5-N | 18.3 | 0.53 | 0.038 | **14** |
| Oct-6-D | 23 | 0.39 | 0.028 | 14 | Oct-6-N | 19.8 | 0.44 | 0.035 | 13 |
| Oct-7-D | 22.6 | 0.47 | 0.029 | 16 | Oct-7-N | 19.1 | 0.46 | 0.036 | 13 |
| Oct-8-D | 17.8 | 0.3 | 0.040 | 7.6 | Oct-8-N | 14.3 | 0.24 | 0.050 | 4.8 |
| Oct-9-D | 18.7 | 0.43 | 0.037 | 12 | Oct-9-N | 12.5 | 0.08 | 0.057 | 1.4 |
| Oct-10-D | 14.4 | 0.19 | 0.050 | 3.8 | Oct-10-N | 15.3 | 0.26 | 0.047 | **5.5** |
| Oct-11-D | 20.1 | 0.1 | 0.034 | 3.0 | Oct-11-N | 16.6 | 0.38 | 0.043 | 8.9 |
| Oct-12-D | 22.5 | 0.27 | 0.029 | 9.4 | Oct-12-N | 17.8 | 1.22 | 0.040 | **31** |
| Oct-13-D | 23.7 | 0.43 | 0.027 | 16 | Oct-13-N | 17.2 | 0.89 | 0.041 | **22** |
| Oct-14-D | 20.1 | 0.37 | 0.034 | 11 | Oct-14-N | 16.3 | 0.49 | 0.044 | 11 |
| Oct-15-D | 22.5 | 0.34 | 0.029 | 12 | Oct-15-N | 19.7 | 0.33 | 0.035 | 9.5 |

150

**Figure S1**. NOAA-HYSPLIT model shown back trajectories for 30 day-night PM$_{2.5}$ sample pairs collected during Sep. 16$^{th}$ to Oct. 15$^{th}$ 2015 from urban center of Beijing, China. Arriving air masses of 500 m above ground level (AGL) were calculated on website of http://ready.arl.noaa.gov/hypub-bin/trajtype.pl?runtype=archive.

[Figure]

155

[Figure]

[Figure]

160

[Figure]

**Figure S2.** $\Delta^{199}$Hg (‰) versus the content of Hg in PM$_{2.5}$ (µg g$^{-1}$) for different subsets of PM$_{2.5}$ samples: a) all data, b) North-West (N-W), c) South-East (S-E) and d) All sunny days (Sun), with Spearman Correlation Coefficient (*R*) and 1-tailed significant (*p*).   The red circles are for daytime samples, while blue circles are for night samples.

[Figure]

170  **Figure S3.** $\Delta^{199}$Hg (‰) versus $\delta^{202}$Hg (‰) for different subsets of PM$_{2.5}$ samples: a) all data, b) North-West (N-W), c) South-East (S-E) and d) All sunny days (Sun), with Spearman Correlation Coefficient (*R*) and 1-tailed significant (*p*).   The red circles are for daytime samples, while the blue circles are for night samples.

[Figure]

175

**Figure S4.** $\Delta^{199}$Hg values of daytime PM$_{2.5}$ samples versus sunshine duration (hr).

[Figure]

180    **Figure S5.** $\Delta^{199}$Hg values of daytime PM$_{2.5}$ samples versus atmospheric ozone content (ppbv).

[Figure]

**Figure S6.** $\Delta^{199}$Hg (‰) versus $\Delta^{200}$Hg (‰) for different subsets of PM$_{2.5}$ samples: a) all data, b) North-West (N-W), c) South-East (S-E) and d) All sunny days (Sun).   The red circles are for daytime samples, while the blue circles are for night samples.   Positive correlations between $\Delta^{199}$Hg and $\Delta^{200}$Hg can be seem in each subsets, with Spearman Correlation Coefficient ($R$) and 1-tailed significant ($p$).

[Figure]

**Figure S7.** $\Delta^{200}$Hg (‰) versus $\delta^{202}$Hg (‰) for different subsets of PM$_{2.5}$ samples: a) all data, b) North-West (N-W), c) South-East (S-E) and d) All sunny days (Sun). The red circles are for daytime samples, while the blue circles are for night samples. Positive correlations between $\Delta^{200}$Hg and $\delta^{202}$Hg can be seem in each subsets with Spearman Correlation Coefficient (*R*) and 1-tailed significant (*p*).

[Figure]